# DeLeaker: Dynamic Inference-Time Reweighting For Semantic Leakage Mitigation in Text-to-Image Models

**Mor Ventura**[1][*]  **Michael Toker**[1][*]  **Or Patashnik**[2]  **Yonatan Belinkov**[1]  **Roi Reichart**[1]

[1]Technion  [2]Tel-Aviv University

{mor.ventura, tok}@campus.technion.ac.il,  orpatashnik@gmail.com
{belinkov, roiri}@technion.ac.il

## Abstract

Text-to-Image (T2I) models have advanced rapidly, yet they remain vulnerable to *semantic leakage*, the unintended transfer of semantically related features between distinct entities. Existing mitigation strategies are often optimization-based or dependent on external inputs. We introduce ***DeLeaker***, a lightweight, optimization-free inference-time approach that mitigates leakage by directly intervening on the model's attention maps. Throughout the diffusion process, *DeLeaker* dynamically reweights attention maps to suppress excessive cross-entity interactions while strengthening the identity of each entity. To support systematic evaluation, we introduce **SLIM** (**S**emantic **L**eakage in **IM**ages), the first dataset dedicated to semantic leakage, comprising 1,130 human-verified samples spanning diverse scenarios, together with a novel automatic evaluation framework. Experiments demonstrate that *DeLeaker* consistently outperforms all baselines, even when they are provided with external information, achieving effective leakage mitigation without compromising fidelity or quality. These results underscore the value of attention control and pave the way for more semantically precise T2I models.[1]

## 1 Introduction

Text-to-Image (T2I) models have shown continuous improvements in image generation capabilities (Ramesh et al., 2021; 2022; Saharia et al., 2022; Black-Forest-Labs, 2024). These advances are largely driven by diffusion-based architectures, which produce high-quality images through iterative denoising (Dhariwal & Nichol, 2021; Ho & Salimans, 2021). Recent state-of-the-art models, such as Diffusion Transformers (DiTs) (Peebles & Xie, 2023), further this progress by adopting transformer-based architectures with uniform global attention, resulting in stronger image–text alignment and improved image quality. Nevertheless, these models remain vulnerable to errors in semantic fidelity, with *semantic leakage* emerging as a particularly persistent challenge.

*Semantic leakage* refers to the unintended transfer of semantically related features between entities in the generated outputs, observed in both image (Rassin et al., 2022; Dahary et al., 2025b) and text generation models (Gonen et al., 2025). An example of this is seen in Fig. 1, where a cow's traits leak into the horse's ears and mouth. Although this phenomenon is a form of a broader problem of image-text misalignment in the context of image generation, it remains highly underexplored.

Prior work employed *layout-based control* to mitigate semantic leakage by assigning entities (e.g., *cow and horse*) to fixed regions (Dahary et al., 2025a;b). While effective in simple scenes, these methods fail in settings that involve interactions between entities (Fig. 1, examples 2–3), where rigid separation is often less natural. By relying on external inputs and bounding-boxes, these methods disregard the model's prior knowledge, overlooking the potential to leverage its internal semantic representations. Moreover, they resort to costly inference-time optimization strategies, commonly used in efforts to refine semantic alignment in T2I models (Chefer et al., 2023; Rassin et al., 2024).

---

[*]Equal contribution.
[1]Code and data are available on the project website at https://venturamor.github.io/DeLeaker.

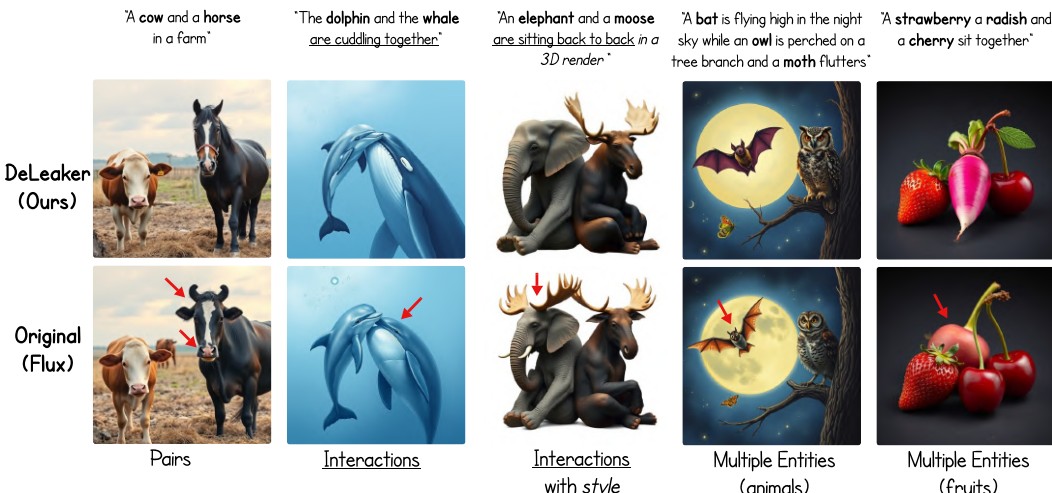

Figure 1: *DeLeaker* **Qualitative Examples**. Top: *DeLeaker* ; Bottom: original outputs. Red arrows mark features affected by semantic leakage. Examples cover five subsets of the SLIM dataset (§3).

In this paper, we introduce **DeLeaker**, a dynamic and lightweight inference-time method designed to mitigate semantic leakage in T2I models. Unlike prior approaches that require costly optimization or external guidance, *DeLeaker* operates by applying synergistic interventions directly to the model's attention mechanism during inference (§2). First, it automatically extracts entity-specific masks from early image-text attention to localize each entity. It then uses these masks to suppress cross-entity leakage by dynamically reducing excessively high attention scores between the different entity regions in both image-text and image-image interactions. Concurrently, it strengthens each entity's *self-identity* by increasing the attention between its corresponding text and image tokens. This targeted reweighting of attention allows *DeLeaker* to mitigate leakage while preserving scene structure and the model's priors. Furthermore, the method remains non-intrusive when no leakage is present.

While developing methods to mitigate semantic leakage is crucial, rigorously evaluating them remains a major hurdle due to the absence of dedicated benchmarks and the limitations of VLM-based automatic evaluation (Dahary et al., 2025a). To address this gap, we introduce a comprehensive evaluation framework, centered around a new dedicated dataset (§3). Our dataset, the *Semantic Leakage in IMages (***SLIM***)* dataset, comprises 1,130 (prompt, seed, image) samples capturing diverse leakage scenarios, including prompts describing visually similar entities, spatial interactions, and multi-entity compositions. SLIM is constructed from a large pool of images generated by the FLUX.1-dev model (Black-Forest-Labs, 2024), using prompts automatically produced by GPT-4o. The images are rigorously filtered through an extensive human filtering process.

Next, we develop an evaluation framework (§4) to assess semantic leakage mitigation. We adopt a *comparative evaluation setup* in which images from before and after the mitigation process are compared. Importantly, our framework breaks down the challenging comparative evaluation into a series of discrete logical steps. The process begins with the identification of differences between entities to detect semantic leakage, followed by the ranking of the mitigation's success. Additionally, we include evaluation of the image-text semantic alignment and the preservation of the original image quality and perceptual similarity. Our automatic evaluation pipeline is validated by an extensive human study (980 responses).

In experiments with FLUX (§6), *DeLeaker* significantly outperforms all evaluated baselines in both automatic and human evaluations. This includes prompt-based baselines and layout-based baselines (§5) that require additional information, typically from external LLMs. To confirm its generalizability, we applied *DeLeaker* to another model, SANA (Xie et al., 2024), and found it to be similarly effective at mitigating semantic leakage. To understand the source of *DeLeaker*'s advantage, our ablation study (§7) reveals that *DeLeaker*'s strength derives from its cross-modal attention interventions, particularly the image-text strengthening that preserves self-identity.

To summarize, our contributions include: **(1)** *DeLeaker*, a dynamic, lightweight inference-time method for mitigating semantic leakage in T2I models while preserving image quality and perceptual similarity, **(2)** the first dedicated dataset explicitly designed to evaluate semantic leakage, and **(3)** an automated evaluation pipeline for large-scale assessment supported by a human study. We hope this work will inspire further research toward more controlled and reliable generative models.

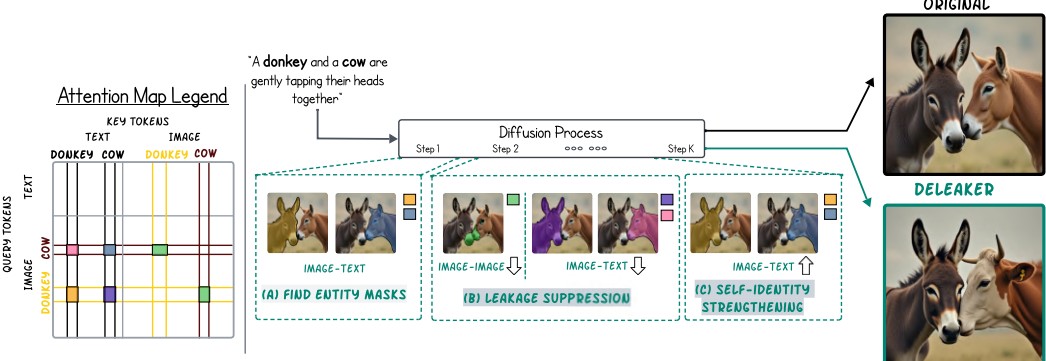

Figure 2: ***DeLeaker*** **Scheme.** Our method applies attention-based interventions during the diffusion process: (A) automatically extracting entity-specific masks from early image-text attention maps; (B) mitigating cross-entity leakage by suppressing attention across entities in both image-text and image-image interactions; and (C) strengthening self-identity by increasing attention from each entity's text tokens to its own image tokens. The attention map legend (left) shows how entities interact, where colors denote different interaction regions. The final output (right) presents the image output with *DeLeaker* compared to the original image, when *DeLeaker* is not applied.

## 2 *DeLeaker*

Our method, *DeLeaker*, aims to mitigate semantic leakage in DiT T2I models. As illustrated in Fig. 2, it relies solely on dynamic reweighting interventions at inference time in the self-attention mechanism during the diffusion process, and consists of three key steps. First, it identifies the image tokens (masks) corresponding to entities, i.e., the regions where they should appear in the generated image (§2.1). Second, it suppresses the connections between entities in both the text-image and image-image self-attention maps (§2.2). Finally, it enhances the self-identity of each entity by strengthening the connection between its text token and the corresponding image tokens (§2.2). In §7, we present an ablation study, which demonstrates the importance of each intervention.

### 2.1 ATTENTION-BASED ENTITY MASKING

To mitigate semantic leakage, we first localize for each textual entity $e_i$ in the prompt, the set of image tokens it governs, and then manipulate the attention maps using these localizations. Specifically, we use the pre-softmax attention scores, $Attn$, between all image tokens, $\mathcal{I}$, (used as queries, $q$) and the set of text tokens, $\mathcal{E}_i^{\text{txt}}$ ,(used as keys, $k$). The corresponding image tokens $\mathcal{E}_i^{\text{img}}$ are selected by averaging attention scores across heads and applying a dynamic threshold based on the mean, $\mu_i$ and standard deviation, $\sigma_i$ of the attention distribution (Eq. 1). Following prior work on UNet-based diffusion (Hertz et al., 2022; Binyamin et al., 2025), we observe that even early diffusion steps yield sufficiently accurate masks (§B.1, Fig. 5). Thus, we aggregate attention maps across early steps in the diffusion process to create a mask for each entity. We apply smoothing techniques on the masks: (1) temporal smoothing by averaging over the accumulated history maps, and (2) spatial smoothing via filtering, resulting in more stable and coherent entity masks (see §B.1, Fig. 6).

$$\mathcal{E}_i^{\text{img}} = \{q \in \mathcal{I} \mid Attn_{qk} > \mu_i + \beta_1 \cdot \sigma_i, k \in (\mathcal{E}_i^{\text{txt}} \cap \mathcal{I})\} \tag{1}$$

## 2.2 ATTENTION-BASED LEAKAGE MITIGATION

**Leakage Suppression.** Utilizing attention-based entity masks, we focus on *cross-entity attention*, which measures how the image tokens of one entity, $e_i$, attend to the text or image tokens of another entity, $e_j$. While cross-entity relations are a primary source of semantic leakage, they are also essential for creating meaningful interactions, such as shared actions and poses. Therefore, our goal is not to eliminate these connections entirely, but to selectively suppress only those causing leakage while preserving beneficial ones. We hypothesize that high attention values in image-image relations (Eq. 2) represent unwanted semantic transfer (akin to high-frequency noise), while lower values reflect desirable, meaningful interactions (the core signal). Specifically, we apply a unified suppression mechanism by zeroing out attention scores (Eq. 3, first two cases). This involves fully suppressing all cross-entity image-text attention scores while also suppressing image-image attention scores that exceed one standard deviation, multiplied by a coefficient, $\beta_2$, above their mean. This intervention is applied only after the initial, attention-based entity masks have been formed.

$$\mathrm{H}_{ij}^{\text{img-img}} = \{(q, k) \mid Attn_{qk} > \mu_{ij} + \beta_2 \cdot \sigma_{ij}, q, k \in \mathcal{I}\} \tag{2}$$

**Strengthening Self-Identity Alignment.** Finally, we introduce a third intervention to strengthen the connection between each entity's text tokens and its corresponding image tokens (Eq. 3, third case). This enhancement improves the *self-identity* of each entity. We apply this by multiplying the relevant attention scores by a coefficient $\alpha > 1$ ($\alpha$ ablations in §B.1, Fig. 7). The coefficients are empirically chosen based on a qualitative review of a few samples external to the SLIM dataset.

$$Attn'_{qk} = \begin{cases} -\infty & \text{if } q \in \mathcal{E}_i^{\text{img}}, k \in \mathcal{E}_j^{\text{img}}, \text{ and } (q, k) \in \mathrm{H}_{ij}^{\text{img-img}} \\ -\infty & \text{if } q \in \mathcal{E}_i^{\text{img}}, k \in \mathcal{E}_j^{\text{txt}} \\ \alpha \cdot Attn_{qk} & \text{if } q \in \mathcal{E}_i^{\text{img}}, k \in \mathcal{E}_i^{\text{txt}} \\ Attn_{qk} & \text{else} \end{cases} \tag{3}$$

Here $Attn_{qk}$ is the single pre-softmax attention score between the tokens $q$ and $k$. The terms $\mu_i$ and $\sigma_i$ are respectively the mean and standard deviation (std) of attention scores for entity $i$'s image tokens. Similarly, $\mu_{ij}$ and $\sigma_{ij}$ are the mean and std for attention between the image tokens of entities $i$ and $j$.

Having established the method, we next turn to the dataset design that enables a systematic evaluation of its effectiveness. See §C for *DeLeaker*'s full equations and hyperparameter values.

## 3 THE SLIM DATASET: SEMANTIC-LEAKAGE IN IMAGES

Prior efforts to mitigate semantic leakage (Dahary et al., 2025b) and improve semantic alignment Feng et al. (2022) have relied on general-purpose benchmarks such as DrawBench (Saharia et al., 2022) and MS-COCO (Lin et al., 2014). These benchmarks, however, do not specifically target semantic leakage. This is because the phenomenon is mainly associated with the visual similarity of entities (Dahary et al., 2025b), a condition that rarely appears in their general-purpose prompts. Consequently, prior work has often drawn conclusions from evaluating extremely small subsets of these datasets, sometimes only a few dozen samples (Chefer et al., 2023; Dahary et al., 2025b).

To fill this gap, we introduce SLIM, which is, to the best of our knowledge, the first dataset explicitly designed to study and evaluate visual semantic leakage at scale. It contains 1,130 samples, each with a prompt, a generation seed, and a corresponding image exhibiting semantic leakage, all generated using FLUX (Black-Forest-Labs, 2024). SLIM is organized into five subsets, as detailed below (examples in Fig. 1), and was curated through a two-step process: large-scale generation followed by human-guided filtering. To validate that *DeLeaker*'s performance is not limited to FLUX, we create an additional test set using SANA (Xie et al., 2024). Due to the extensive data filtering required, this supplementary set contains 370 samples.

**Large-Scale Generation & Dataset Design.** Building on the finding that semantic leakage is associated with visual similarity (Dahary et al., 2025b), we find the effect is particularly acute within fine-grained categories (e.g., dog breeds). Motivated by this, we focus on animals and fruits for controlled evaluation. Starting from a curated list of 90 animals (Banerjee, 2023), we use GPT-4o

to expand it to 200 animals and generate 200 descriptive prompts, each pairing visually similar animals. We then produce corresponding images using five seeds per prompt. We leverage the animal pairs subset to create increasingly complex scene configurations, hypothesized to be associated with stronger leakage (see Table 3), including interactions (e.g., hugging), shared visual styles (e.g., comics), and multiple entities (triplets). To probe semantic leakage in a different domain, we similarly expand our dataset to include a fruits & vegetables subset based on an existing list of 36 fruits (Seth, 2019), where leakage is rare in pairs but emerges in triplets. Notably, subsets with multiple entities tend to present challenges beyond semantic leakage, as they are also prone to *entity count errors* (i.e., missing or added entities).

**Human-Guided Filtering of Semantic Leakage.**   We filter the large-scale set to include only images that exhibit detectable semantic leakage. This is achieved through a two-stage process: an initial large-scale filtering using a noisy automatic pipeline, followed by a second round of manual verification through human annotation.[2] We designed a rigorous structured human annotation protocol for detecting semantic leakage, detailed in §F.1. See §G for subset sizes through the filtering and prompt examples.

## 4 EVALUATION

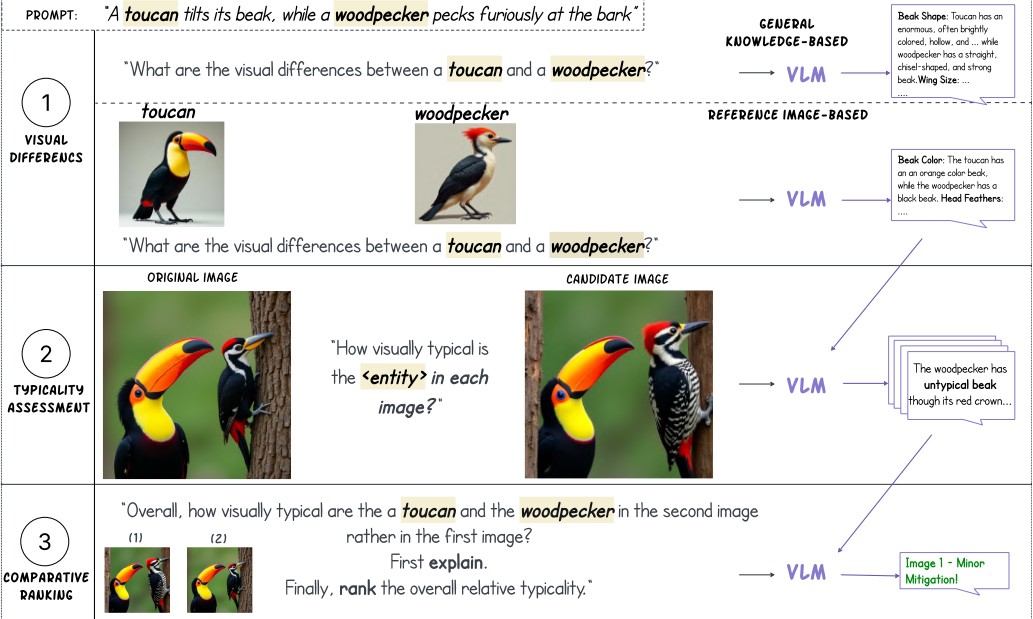

Figure 3: **Our Automatic Evaluation Framework for Assessing Semantic Leakage Mitigation.** The framework consists of three main steps: (1) visual difference extraction, (2) typicality assessment, and (3) comparative ranking. Step 1 is divided into two parts: one based solely on the input prompt (top) and the other employing reference images generated for each entity (bottom). The VLM generates and then merges two independent descriptions into a unified description. Step 2 consists of four typicality questions, one for each entity in each image, guided by the unified differences identified in Step 1. Step 3 employs the outputs of Step 2 to compare both images. It produces a classification indicating the preferred image (Image 1 or Image 2) and the magnitude of change (minor or major).

Evaluating semantic leakage mitigation in T2I models is a major challenge. Prior efforts have often relied on general-purpose metrics (e.g., CLIP score (Radford et al., 2021)) or qualitative judgments, which lack the specificity required for systematic analysis and are often insensitive to subtle, fine-grained errors that characterize semantic leakage (Dahary et al., 2025a). To address this, we introduce a novel automatic evaluation framework centered on a *comparative setup*, which directly contrasts a

---

[2]Specifically, two authors of this paper manually reviewed the images.

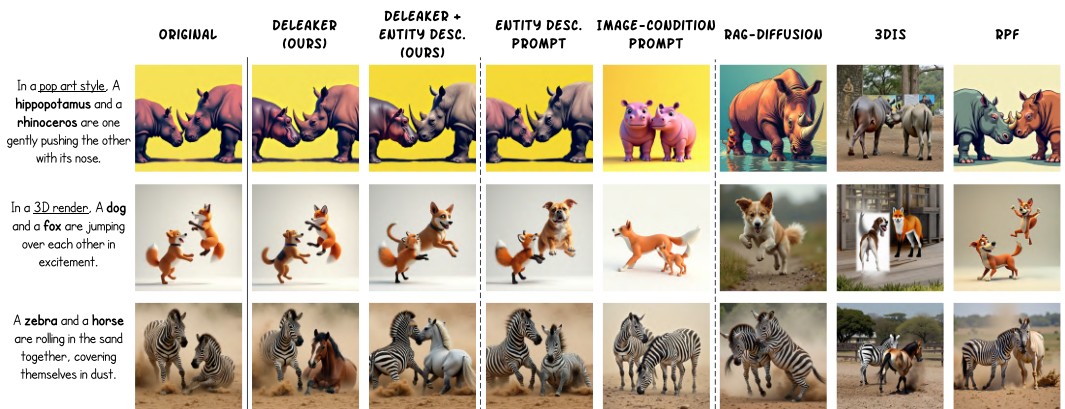

Figure 4: Qualitative comparison across baselines (columns) and three examples (rows).

candidate (mitigated) image against the original. Automating comparative analysis, however, is non-trivial due to limitations in the visual modality of state-of-the-art VLMs (see §B.2). To overcome this challenge, our evaluation pipeline decomposes the complex visual comparison into discrete logical steps, thereby leveraging the more robust reasoning capabilities of the text modality in VLMs (Ventura et al., 2024a; Nikankin et al., 2025). To ensure its reliability, the framework is validated against extensive human assessments. Our evaluation covers two critical dimensions: *leakage mitigation*, which measures the reduction of cross-entity interference, and *preservation*, described next.

**Automatic Evaluation for Leakage Mitigation.** Our framework decomposes the evaluation into three interpretable steps, performed by an external VLM (Gemini 1.5; Team et al. 2024a), as shown in Fig. 3. The process requires four inputs: (1) a prompt, (2) an original image exhibiting semantic leakage, generated by the base model $M$, $I_M^{\text{orig}}$; (3) a candidate image, $I_{M^*}^{\text{cand}}$, generated by a model $M^*$ as a corrected version of the original image; and (4) reference images, generated independently by M, $REF_M^{e_i}$. The reference images act as auxiliary cues, ensuring the evaluation isolates leakage effects rather than the information encoded in the VLM about each of the entities.

The pipeline first identifies **key visual differences** between the entities by combining the VLM's general knowledge with the specific insights from the reference images. Second, it assesses the **typicality** of each entity in both the original and candidate images, measuring how well each matches its expected appearance, based on the key visual differences. Finally, it performs a **comparative judgment** to determine which image better preserves the distinct identity of all entities. To mitigate sensitivity to image order in VLMs (Ventura et al., 2024a), the images are presented in a random order. The evaluation's output is a single discrete label, $c = \Delta(I_{M^*}^{\text{cand}}, I_M^{\text{orig}})$, which represents the change between the two images. This label combines the change's direction (improvement/degradation) and magnitude (major/minor), along with a 'no change' option, resulting in five possible outcomes.

**Preservation Metrics.** In addition to leakage mitigation, we evaluate three preservation aspects: alignment with the original prompt (VQAScore (Lin et al., 2024)), image quality (KID (Jayasumana et al., 2024)), and perceptual similarity to the original image (LPIPS (Zhang et al., 2018)).

**Human Assessment of Leakage Mitigation (User Study).** To establish human baseline preferences and validate the reliability of our automatic evaluation, we conduct a user study on Amazon Mechanical Turk (AMT), resulting in a total of 980 individual responses (see AMT questionnaire in §F.2). Since evaluating leakage with multiple entities is confounded by the difficulty of assessing each entity pair separately, we focus our evaluation on the pair subsets of the SLIM dataset. We randomly sample 60 prompts from these subsets, ensuring equal distribution across the subsets. Each task presents a candidate image generated by one of six baselines representing a range of methods, the original image, and two reference images (one per entity), following the structure of the automatic pipeline. The questionnaire includes two questions that assess the typicality of each entity using a five-point scale aligned with the automatic evaluation. Each task is completed by three annotators, and responses are combined via majority vote. The inter-annotator agreement is 0.52 (quadratic

Table 1: **Automatic and Human Assessment Scores of Semantic Leakage Mitigation.** We compare our method, *DeLeaker* (bottom rows), against layout-based (top) and prompt-based (middle) baselines. The main scores, summarized by a stacked bar visualization, represent the percentage of samples labeled as Mitigation (Major/Minor), No Change, or Degradation (Major/Minor), where a wider portion indicates better performance. The automatic scores are calculated on the SLIM pair subsets (840 samples), while the human scores are gathered from the user study of 60 random samples from that same subset. These are presented alongside preservation metrics (VQAScore, LPIPS, and KID ($\cdot 10^{-2}$)). Arrows ($\uparrow/\downarrow$) indicate the desired direction for improvement.

| Method | Semantic Leakage (Automatic) | | | | | | (Human) | Preservation | | |
| | Visualization | Mitigation $\uparrow$ | | *No Change* | Degradation $\downarrow$ | | Visualization | VQAScore $\uparrow$ | LPIPS $\downarrow$ | KID $\downarrow$ |
| | | Major | Minor | | Minor | Major | | | | |
| RAG-Diffusion | | 17.55% | 4.17% | 5.03% | 8.34% | 64.91% | | 0.42 | 0.72 | 0.09 |
| RPF | | 20.74% | 9.06% | 16.57% | 15.26% | 38.38% | | 0.63 | 0.64 | 0.53 |
| 3DIS | | 29.08% | 8.10% | 7.63% | 10.13% | 45.05% | | 0.62 | 0.76 | 0.96 |
| QwenFLUX | | 17.28% | 7.51% | 15.85% | 12.75% | 46.60% | | 0.49 | 0.61 | 0.46 |
| Instruction Prompt | | 23.92% | 11.54% | 35.35% | 9.28% | 19.88% | — | 0.64 | 0.33 | 0.00 |
| Entity Description Prompt | | 35.60% | 11.07% | 25.71% | 9.17% | 18.45% | | 0.62 | 0.41 | 0.00 |
| DeLeaker | | 46.07% | 9.76% | 25.36% | 5.83% | 12.98% | | 0.68 | 0.22 | 0.00 |
| DeLeaker + Description | | 53.57% | 8.57% | 15.95% | 6.55% | 15.36% | — | 0.65 | 0.43 | 0.01 |

weighted Fleiss' $\kappa$), which validates the correlation between human judgments and our automatic evaluation (Spearman's $\rho$=0.432) as a meaningful proxy. We observe a difference in model-human sensitivity: both typically agree on the change's direction (mitigation vs. degradation) but differ on its magnitude (minor vs. major).

## 5 EXPERIMENTAL SETUP

**Base DiT T2I Models.** We primarily experiment with the state-of-the-art open-source DiT T2I model FLUX.1-DEV (Black-Forest-Labs, 2024), while also applying *DeLeaker* to SANA (Xie et al., 2024) to validate our findings. Unlike earlier UNet-based models (Ronneberger et al., 2015) such as Stable Diffusion (Rombach et al., 2022), where textual information is injected through spatial cross-attention layers at multiple resolutions during denoising, DiTs employ a transformer-based backbone (Vaswani et al., 2017) that processes image and text tokens jointly. This architectural shift promotes capturing complex cross-modal dependencies and achieving more consistent global semantics. The differing text encoders and attention mechanisms in FLUX and SANA are relevant for studying semantic leakage, as these components control how unintended information propagates between modalities. To the best of our knowledge, this setup represents the first exploration of semantic leakage in DiT T2I models. For brevity, the following setup focuses on FLUX, while the full experimental details for SANA are available in §E.1.

**Baselines.** We evaluate *DeLeaker* against *layout-based* and *prompt-based* baselines. Layout-based methods provide explicit priors on image structure to improve compositional control (Chen et al., 2024a;b), making them relevant for semantic leakage as their structure reduces content mixing. Additionally, we include several zero-shot, prompt-based baselines, which are common for improving image-text alignment (Yang et al., 2024). To maintain a fair comparison, all methods are built upon the FLUX base model, as our SLIM dataset is created using FLUX-generated images.

For **layout-based baselines**, we utilized FLUX-based parallel implementations of an existing UNet baseline (Dahary et al., 2025b), specifically *RPF* (Chen et al., 2024a), *RAG-Diffusion* (Chen et al., 2024b), and *3DIS* (Zhou et al., 2025). These baselines differ in their inputs and conditioning strategies. The first, *RPF*, leverages regional prompts within bounding boxes while eliminating cross-bounding-box attention. The second, RAG-Diffusion, constrains self-attention to local text descriptions within each box, but only during the initial steps of the diffusion process. Finally, *3DIS* conditions on bounding boxes to generate a depth map as an additional input. It is important to note that all three baselines rely on external LLMs or additional models as guidance (§H.1).

For **prompt-based** baselines, we employ three methods. The first is an implicit instruction to generate an image without semantic leakage between entities, referred to as the *Instruction Prompt*. Since

Table 2: **_DeLeaker_ Ablation Study.** Configurations are divided into two types: (1) `W/O` rows (top four) represent the removal/addition of a specific component, while (2) `Only` rows (bottom three) isolate each component independently. Ratios are reported relative to the full _DeLeaker_ scores baseline, with values closer to 1.0 indicating similarity. Darker hues indicate stronger contributions, color-coded as  positive  and  negative . Signs indicate attention suppression (-) or strengthening (+).

| Configuration | Leakage Mitigation (Relative to DeLeaker) | | | | |
| --- | --- | --- | --- | --- | --- |
| | _Improvement ↑_ | | _No Change_ | _Degradation ↓_ | |
| | Major | Minor | | Minor | Major |
| DeLeaker | 1.00 | 1.00 | 1.00 | 1.00 | 1.00 |
| `W/O` Image-Image (-) | 1.01 | 1.04 | 1.05 | 0.73 | 0.97 |
| `W/O` Image-Text (-) | 0.93 | 0.78 | 1.10 | 1.04 | 1.18 |
| `W/O` Image-Text (+) | 0.54 | 0.82 | 1.73 | 1.20 | 1.24 |
| `With` Text-Text (-) | 0.91 | 0.91 | 1.08 | 1.20 | 1.16 |
| `Only` Image-Image (-) | 0.26 | 0.61 | 2.44 | 1.35 | 0.96 |
| `Only` Image-Text (-) | 0.54 | 0.88 | 1.88 | 1.00 | 0.99 |
| `Only` Image-Text (+) | 0.90 | 0.99 | 1.23 | 0.88 | 0.96 |

T2I models are not trained for instruction-following, we also experiment with explicitly describing each entity and its appearance, referred to as the _Entity Description Prompt_. To illustrate, the prompt _"A zebra and a horse are riding in the sand together..."_ (Fig. 4) is enriched with LLM-generated entity attributes, such as _"the zebra has dense black-and-white stripes, while the horse has white fur and a blond tail.."_. The final method is the _Image-Conditioned Instruction Prompt_, where the model (Qwen2VL-Flux; Lu 2024) is instructed to mitigate leakage based on the original image.

## 6 RESULTS

Table 1 presents the automatic and human evaluation of leakage mitigation across all baselines on the SLIM pair subsets, and Fig. 4 presents qualitative examples (see additional examples in §D). Complementary results are in §E, including SANA's scores and results with multiple entities.

**_DeLeaker_ outperforms baselines in mitigating semantic leakage.** Our automatic evaluation shows that _DeLeaker_ achieves the highest rate of semantic leakage mitigation with minimal degradation. Human evaluation strongly confirms these findings, with annotators judging that _DeLeaker_ improved the image in a clear majority of cases (67.8% total improvement), outperforming all other methods. Furthermore, adding entity descriptions to _DeLeaker_ (similar to the 'Entity Description Prompt' baseline) offers only minor gains, indicating that _DeLeaker_ is highly effective on its own. Among the other baselines, the text prompt-based methods have a combined degradation rate of just 24.2%, which is significantly lower than the rates for layout-based methods, all of which are over 50%.

**_DeLeaker_ preserves fidelity and quality.** Beyond leakage mitigation, _DeLeaker_ excels at preserving image fidelity and quality. It achieves the lowest LPIPS score (0.22), meaning it best preserves the original image, which indicates that the method effectively leverages the model's internal knowledge and priors, applying only minimal, necessary interventions. _DeLeaker_ also attains the highest VQAScore (0.68), signifying strong image-text alignment. Moreover, it achieves the lowest KID score (0.00) alongside the prompt-based baselines, demonstrating that strong leakage reduction is achieved without sacrificing original image quality. Notably, when applied to images without leakage (§D, Fig. 12), _DeLeaker_ induces negligible changes, thereby remaining non-intrusive.

## 7 ABLATION STUDY & ANALYSIS

Table 2 presents an ablation study assessing the contribution of each _DeLeaker_ component. It includes two configurations: (1) `W/O` ablations, where components are removed from or added to the full method while the others are applied, and (2) `Only` ablations, where components are tested in isolation. Results are reported as ratios relative to the full automatic leakage mitigation scores of _DeLeaker_.

**The most influential intervention is** *self-identity (image-text) strengthening*. When applied alone, it achieves a 0.90 ratio in the "major improvement" (leftmost column). Conversely, when removed, the original score drops by 46% (to 0.54), confirming its key role in leakage prevention. The second most influential intervention is cross-entity image-text suppression. Omitting it causes a 29% reduction in improvement (major and minor). Furthermore, when applied in isolation, it accounts for 0.54 (second-to-last row) of the total major improvement with almost no degradation, demonstrating its significant contribution. While cross-modality interventions are found to be effective, **self-modality interventions have only a limited impact**. Suppressing text-text interactions degrades performance by 9% to 20%, suggesting that leakage in DiT T2I models is primarily due to cross-modal misalignment. Similarly, weakening image-image interactions has a small and inconsistent impact (see absolute values in §E.3). Taken together, our analysis pinpoints the root of semantic leakage not to weaknesses within each modality, but to the faulty alignment between them, suggesting a promising direction for future research.

Finally, we analyze mitigation performance across the SLIM subsets. As shown in Table 3, the rate of successful mitigation increases dramatically with subset complexity. The total improvement rate (major and minor) rises from 42.4% for simple Animal Pairs to 62.6% for Animal Interactions, and further to 66.4% for the most complex Animal Interactions + Style subset. This provides clear evidence for our hypothesis: **more complex prompts elicit stronger semantic leakage**. This validates their use in SLIM as stress tests for semantic leakage.

Table 3: **SLIM Subset Analysis with *DeLeaker*.**

| Subset | Visualization |
|---|---|
| Animal Pairs | |
| Animal Interactions | |
| Animal Interactions + Style | |

## 8 RELATED WORK

**Alignment in T2I models.** Ensuring alignment between the text prompt and the generated image is a fundamental objective in T2I models, serving both as a generation condition and as an evaluation goal (Xie et al., 2019; Hu et al., 2023; Yarom et al., 2024; Gordon et al., 2023). Many approaches rely on encoding-based methods, such as joint image-text embeddings (e.g., CLIP), which were found to be ineffective for fine-grained details between modalities (Liang et al., 2022; Yuksekgonul et al., 2022; Koishigarina et al., 2025) (see §B.2). While recent work has employed VLMs as alignment evaluators (Li et al., 2023), they are unsuitable for detecting semantic leakage. VLMs struggle with the fine-grained details (Tong et al., 2024; Yu et al., 2025) and complex reasoning required for multi-image comparisons (Ventura et al., 2024b). This means a direct approach is insufficient, highlighting the need for a more guided, step-by-step evaluation process. To the best of our knowledge, no evaluation method explicitly targets semantic leakage, despite its prevalence in T2I models (see §G.1, Table 17). Addressing this gap is a central focus of our work, in which we introduce a dedicated method to mitigate semantic leakage and a corresponding evaluation framework.

**Semantic Leakage in T2I models.** Leakage in T2I models was first identified by Rassin et al. (2022) in UNet-based T2I models, though a direct mitigation was not proposed. While subsequent research has addressed related visual artifacts such as attribute binding (see §A for a distinction from semantic leakage; Feng et al., 2022; Rassin et al., 2024), composition errors, and missing entities (Binyamin et al., 2025) by modifying the attention mechanism, these works do not directly address semantic leakage. To the best of our knowledge, Dahary et al. (2025a;b) were the only ones to explicitly tackle this problem. However, their solutions rely on external layout guidance or costly optimization. In contrast, we introduce *DeLeaker*, a lightweight, training-free, guidance-free semantic leakage mitigation method.

**Semantic Leakage in Language Models.** Semantic leakage has only recently been recognized as an issue in state-of-the-art language models like GPT-4o, where prompt information unintentionally biases the output (Gonen et al., 2025). While progress has been made in diagnosing semantic leakage, with one cause identified as leakage between lexical items in the text encoder (Kaplan et al., 2025), effective mitigation remains an open problem. Therefore, our work focuses on developing a novel mitigation strategy while also investigating the origins of leakage through our method's ablations.

## 9 CONCLUSIONS

This work introduces *DeLeaker*, a lightweight inference-time approach that effectively mitigates semantic leakage in DiT-based T2I models without relying on external information such as bounding boxes. By directly modulating attention patterns during inference, *DeLeaker* mitigates leakage while preserving image-text alignment and image quality. It outperforms existing baselines across diverse scenarios. Complemented by the first dedicated SLIM dataset and comparative evaluation framework, this work provides both a practical solution and a comprehensive foundation for a systematic study of semantic leakage in T2I models.

Future research could expand the SLIM dataset into new domains to explore cross-domain leakage scenarios. Furthermore, SLIM could be used to train leakage classifiers or, when paired with *DeLeaker* outputs, to fine-tune models to inherently avoid semantic leakage. While *DeLeaker* specifically targets T2I models, extending our work to address semantic leakage in other modalities, such as 3D or video, is a natural next step. We hope this work stimulates further progress on new methods, systematic evaluations, and dedicated datasets to address key problems in T2I generation.

## REPRODUCIBILITY STATEMENT

To ensure reproducibility, all code and the newly introduced SLIM dataset will be made publicly available. Our experiments are based on open-source T2I models, FLUX.1-dev and SANA, with all baselines and their configurations clearly documented in §H.1. Key hyperparameters for *DeLeaker*, such as attention reweighting coefficients and the specific diffusion step ranges for interventions, are detailed in §C, Table 6. Moreover, our automated evaluation framework is thoroughly described, with the exact VLM prompts provided in §F to allow for complete replication of our evaluation process.

## ETHICS STATEMENT

In this work, we utilized AI models for several tasks. For grammar improvement, we used Gemini 2.5 Pro. For code completion, we used Claude 4 Sonnet. In all instances, every suggestion or line of code generated by a model was carefully reviewed by the authors to ensure it aligned with our original intentions before being accepted. Finally, as detailed in §3 and §F, we also used LLMs and VLMs for data creation and evaluation.

## ACKNOWLEDGEMENTS

Mor Ventura and Roi Reichart were partially supported by the MOST grant on Advancing Human-Context Reasoning in "Vision-Language Models". Michael Toker is supported by the Azrieli Graduate Studies Fellowship. Yonatan Belinkov was funded by Coefficient Giving, the Israel Science Foundation (grant No. 2942/25), and the European Union (ERC, Control-LM, 101165402).

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

## A  SEMANTIC LEAKAGE: CONCEPTUAL CLARIFICATION AND SCOPE

### A.1  DISTINCTION FROM ATTRIBUTE BINDING

Semantic leakage and attribute binding (Rassin et al., 2024) represent two related but distinct challenges in T2I generation (see Table 4). **Attribute binding** refers to the failure to correctly associate explicitly mentioned attributes with their intended entities in the input prompt. For instance, in prompts such as "a yellow flamingo and a pink sunflower" or "a red frog and a blue rabbit", models may misplace attributes (e.g., rendering the rabbit as red or the frog as blue), resulting in incorrect color-to-entity assignments. The source of the error is thus a misalignment between the linguistic specification of attributes and their grounding in the image.

In contrast, **semantic leakage** arises not from the wrong binding of attributes explicitly stated in text, but from the unintended transfer of semantically related features between entities. This phenomenon is primarily driven by the visual similarity of the entities, making it more likely to occur between a horse and a donkey than between a cow and a parrot. On the other hand, attribute binding can also occur between visually dissimilar entities (e.g., "a red cow and a white parrot"). Here, features attend their semantically similar counterparts across entities, for example, the ears of one animal influencing the ears of another, or the shape of a mouth blending between two species. This leads to cross-entity entanglement of features that are not even explicitly mentioned in the textual prompt, but emerge due to the semantic proximity of visual parts (e.g., cow ears leaking into a horse's ears).

Table 4: Comparison between *Attribute Binding* and *Semantic Leakage*.

| Aspect | Attribute Binding | Semantic Leakage |
|---|---|---|
| **Definition** | Misalignment between textual attributes and their intended entities. | Unintended transfer of semantically related features between entities. |
| **Source** | Explicit attributes in the text prompt (e.g., colors, shapes). | Implicit similarity between visual features (e.g., ears, eyes, mouths). |
| **Primary Cause** | Confusion over explicit attributes, regardless of entity similarity (e.g., "a red cow and a white parrot"). | Visual/semantic proximity of the entities themselves (e.g., more likely between a horse and a donkey). |
| **Example Prompt** | "A yellow flamingo with a pink sunflower"; "A red frog and a blue rabbit." | "A cow and a horse in a farm." |
| **Error Manifestation** | Attributes swapped or misplaced (e.g., a blue frog instead of a red frog). | Feature entanglement across entities (e.g., cow traits appearing in the horse's ears). |
| **Commonality** | Both result in semantically inconsistent outputs that reduce fidelity to the intended meaning. | |

### A.2  DIFFERENTIATION FROM LEAKAGE IN IMAGE-TO-IMAGE GENERATION

More recently, leakage has also been discussed in the context of style-content entanglement in image-to-image generation using reference images (Frenkel et al., 2024; Li et al., 2025). This line of work, however, focuses on a different type of leakage that occurs between style and content, rather than on the internal semantic leakage between entities within the same image, which is the focus of our study. Image-to-image editing frameworks offer another possible direction for addressing this challenge. However, they involve computationally expensive double inference and rely on external inputs, prompt optimization (Yang et al., 2023) or adapters optimizations often resulting in identity preservation issues (Slobodkin et al., 2025). In contrast, our method is both training-free and guidance-free. It achieves high semantic consistency with the original image without requiring prior generation or post hoc correction.

# B    FURTHER ANALYSIS & ABLATIONS

## B.1    *DeLeaker* COMPONENTS ABLATIONS

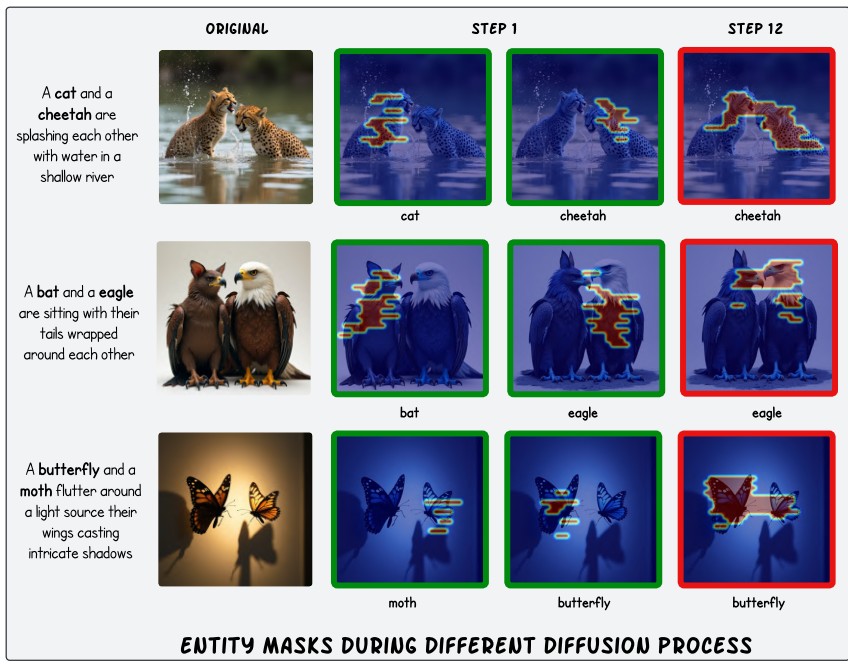

Figure 5: **Entity masks are accurate even in the first diffusion step (50 blocks; green frame).** This is particularly evident in semantic leakage cases, where these initially clear masks begin to blend by a middle step (660 blocks; red frame). The full process consists of 20 diffusion steps (1140 blocks total).

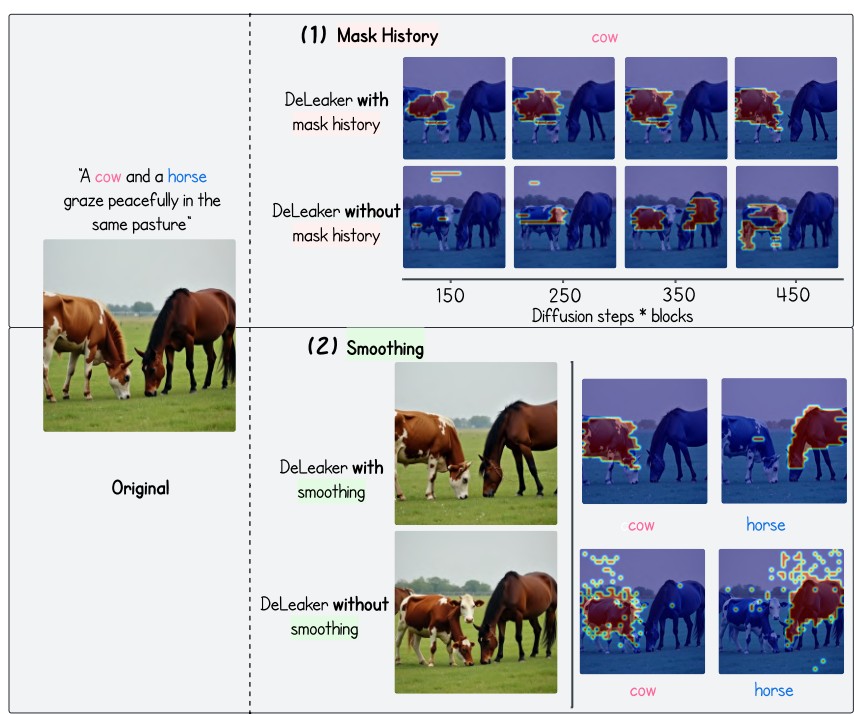

Figure 6: **Ablation study of *DeLeaker*'s smoothing techniques on entity masks.** The figure demonstrates the impact of two components: (**Top**) temporal smoothing and (**Bottom**) spatial smoothing.

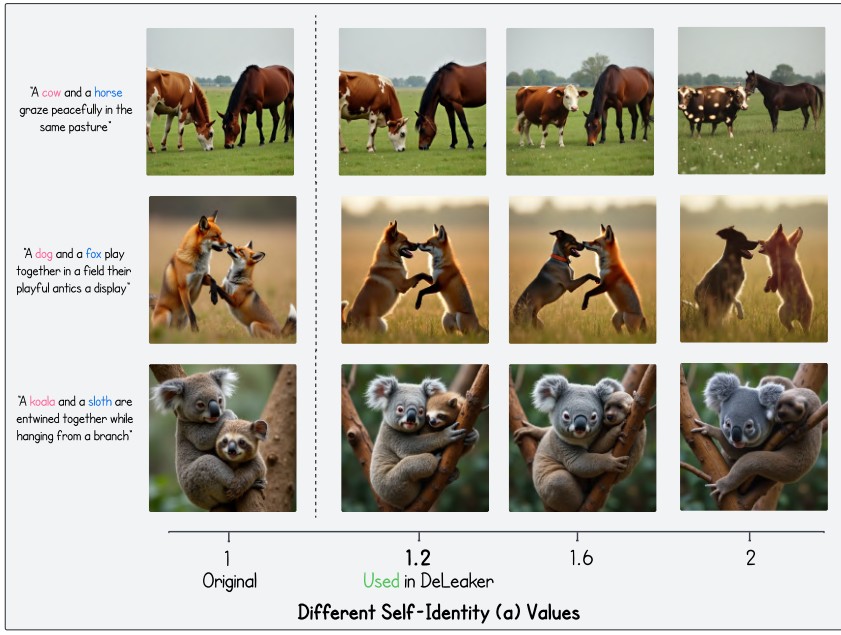

Figure 7: **Effect of varying the self-identity strengthening coefficient ($\alpha$) in *DeLeaker*.** Multiplying the image-text representation by $\alpha$ helps mitigate semantic leakage. This coefficient was empirically optimized on a small set of images, where we found $\alpha = 1.2$ effectively mitigates semantic leakage. Whereas, higher values, such as $\alpha = 2.0$, introduces visual artifacts.

**Analysis: Attention Differences between *DeLeaker* and Original** To further analyze the contribution of *DeLeaker*'s cross-entity components (image-image and image-text), we track **the progression of semantic leakage across model (FLUX-dev) blocks and diffusion steps**. We compute the average proportion of tokens attending to the other entity, exceeding the dynamic leakage threshold, relative to the number of tokens in the entity mask. For each entity pair, we measure leakage in both directions and take the maximum, as leakage typically occurs in only one direction ($e_i \rightarrow e_j$). The analysis is performed under two conditions: standard inference (original) and inference with *DeLeaker*. Figure 8 shows the relative mean difference in leakage progression between the two settings. While the image-image component's effect is bounded at a high value, partially explaining its smaller apparent change, the data still suggests this intervention has a lower impact on mitigating semantic leakage.

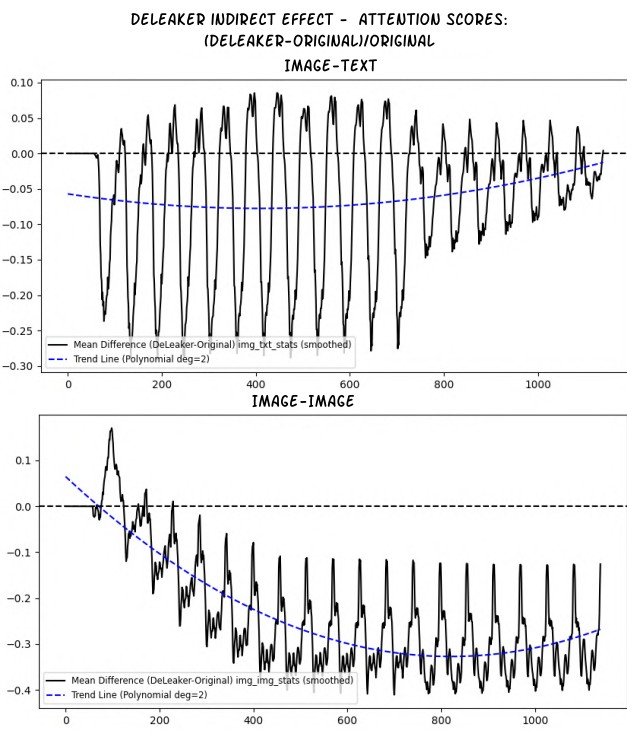

Figure 8: **Analysis of Leakage Mitigation Progression.** The figure shows how *DeLeaker*'s cross-entity components mitigate semantic leakage throughout the FLUX diffusion process (steps × blocks). The y-axis represents the relative change in cross-entity attention between the *DeLeaker* run and the original run. The top and bottom plots show the effects for the image-text and image-image components, respectively.

### B.2 AUTOMATIC EVALUATION BASED ON PREVIOUS EFFORTS

Evaluating the success of semantic leakage mitigation fundamentally requires a comparative analysis between the original and the corrected image. Automating this comparison is non-trivial, however, as state-of-the-art methods suffer from critical limitations. Vision-Language Models (VLMs), for instance, exhibit order sensitivity where their judgment is biased by image presentation order, and possess unreliable visual encodings that fail in zero-shot comparisons (Ventura et al., 2024a). Similarly, joint-encoding models like CLIP are unreliable due to significant cross-modal alignment gaps, often failing to correctly match text with visual information (Liang et al., 2022). These limitations highlight the need for a more robust, step-by-step evaluation pipeline, as simple proxies are insufficient for this nuanced task.

We investigated whether standard metrics from joint-encoding models like CLIP and BLIP could serve as a proxy for our evaluation pipeline. To test this, we examined two conditions for both models: a self-identity check, which compares an entity's image crop with its own name (e.g., a

horse image vs. the text *"horse"*, and a **cross-entity** check, which compares the image crop with the other entity's name (e.g., a horse image vs. the text *"cow"*). With CLIP, we measured the direct image-text similarity score. With BLIP, we queried the model with a question (e.g., *"Is this a horse in the image?"*) and used the predicted probability of the answer being *"Yes"*.

We then performed a Spearman's rank correlation analysis between these CLIP and BLIP-based scores and our automatic evaluation labels (major improvement, minor improvement, no change, minor degredation, major degredation). The analysis was conducted on our 821-sample pair subset, using our automatic labels as the ground truth, which themselves correlate moderately with human judgments.

The results, as presented in Table 5, show **no statistically significant correlation** across all tested metrics. The correlation coefficients were found to be negligible, ranging from approximately $-0.04$ to $0.03$, with all corresponding p-values being high ($p \gg 0.05$). This demonstrates that simple, off-the-shelf CLIP and BLIP-based measurements fail to capture the nuances of semantic leakage, reinforcing the need for our structured, multi-step evaluation pipeline.

Table 5: Spearman's rank correlation ($\rho$) between our automatic evaluation labels and various metrics derived from CLIP and BLIP ($N = 821$). In all cases, the correlation is statistically insignificant ($p \gg 0.05$).

| Model | Metric Type | Spearman's $\rho$ | p-value |
|-------|-------------|-------------------|---------|
| CLIP | Self-Identity | 0.010 | 0.773 |
| | Cross-Entity | $-0.010$ | 0.773 |
| BLIP | Self-Identity | 0.027 | 0.440 |
| | Cross-Entity | $-0.027$ | 0.440 |

# C  *DeLeaker* METHOD: COMPLEMENTARY DETAILS

Table 6: Technical details of *DeLeaker*.

| Parameter | Parameter Group | Value | Goal |
|---|---|---|---|
| **General T2I Parameters** | Number of inference steps | 20 | |
| | Guidance scale | 3.5 | |
| ***DeLeaker*-Specific Parameters** | $\alpha$ | 1.2 | self-identity strengthening |
| | $\beta_1$: Std. coefficient (text-image) | 0.9 | entity mask |
| | $\beta_2$: Std. coefficient (image-image) | 2 | image-image suppression |
| | $t_{\text{start-aggregation}}$ | 12 | diffusion step of start aggregating entity masks |
| | $t_{\text{end-aggregation}}$ | 456 | diffusion step of stop aggregating entity masks |
| | $t_{\text{start-intervention}}$ | 57 | diffusion step of start interventions (suppression and strengthening) |
| | $t_{\text{end-intervention}}$ | 741 | diffusion step of stop interventions (suppression and strengthening) |

## C.1  *DeLeaker* IMPLEMENTATION: TECHNICAL DETAILS

**Entity Mask**  The first step of *DeLeaker* is to find and extract the relevant image tokens of each entity in the prompt. We find, similarly to previous work in UNet-based diffusion models (Binyamin et al., 2025), that early diffusion steps yield more accurate entity segmentation masks compared to later ones. Surprisingly, even within a single partial diffusion step, this method produces reliable results. Based on this observation, we aggregate attention maps for each entity across selected diffusion blocks and timesteps. Specifically, from $t_{12}$ to $t_{171}$, where one diffusion step is consists of 57 blocks, while we run on 20 diffusion steps (results in total 1140 blocks in the diffusion process).

Due to significant variation across blocks and timesteps, we apply two smoothing techniques to improve mask quality: (1) Spatial smoothing: applying a smoothing filter to fill small holes and remove isolated artifacts. In this refinement, we apply several filters. The first is a **morphological closing** operation which fills small holes within the predicted masks. Then, we apply a **morphological opening** to eliminate spurious noise pixels, both using a $3 \times 3$ elliptical structuring element. (2) Temporal smoothing (History): we enforce **temporal coherence** by averaging the attention-based masks across a constrained window of subsequent transformer blocks and time steps. This window deliberately excludes the initial block of the first time-step. These that are very noisy and limited in duration to prevent the erroneous merging of distinct object masks over time. The combined methodology yields masks that are both spatially clean and temporally stable. Together, these steps produce cleaner and more consistent segmentation masks (see Figures in §B.1).

**SANA-based *DeLeaker***  We found that the image-image component yielded inconsistent results; while it sometimes improved leakage mitigation, it also occasionally introduced visual artifacts. Due to this unpredictable behavior, we excluded it from the final SANA configuration.

## C.2  FULL MATHEMATICAL FORMULATION

The standard scaled dot-product attention mechanism is calculated as:

$$\text{Attention}(Q, K, V) = \text{softmax}\left(\frac{QK^T}{\sqrt{d_k}}\right) V \tag{4}$$

where $Q, K, V$ are the Query, Key, and Value matrices, and $d_k$ is the dimension of the keys. The term $\text{Att} = QK^T$ represents the raw, unnormalized similarity scores before scaling and the softmax operation. The following sections detail a process for modifying these raw scores.

**Find Entity Masks**

$$\mu_i = \frac{1}{|\mathcal{I}||\mathcal{E}_i^{\text{txt}}|} \sum_{q \in \mathcal{I}} \sum_{k \in \mathcal{E}_i^{\text{txt}}} \text{Att}_{qk} \tag{5}$$

$$\sigma_i = \sqrt{\frac{1}{|\mathcal{I}||\mathcal{E}_i^{\text{txt}}|} \sum_{q \in \mathcal{I}} \sum_{k \in \mathcal{E}_i^{\text{txt}}} (\text{Att}_{qk} - \mu_i)^2} \tag{6}$$

$$\mathcal{E}_i^{\text{img}} = \{q \mid \text{Att}_{qk} > \mu_i + \beta_1 \cdot \sigma_i, k \in \mathcal{E}_i^{\text{txt}}, k \in \mathcal{I}, q \in \mathcal{I}\} \tag{7}$$

**Modify Attention Scores**

$$\mu_{ij} = \frac{1}{|\mathcal{E}_i^{\text{img}}||\mathcal{E}_j^{\text{img}}|} \sum_{q \in \mathcal{E}_i^{\text{img}}} \sum_{k \in \mathcal{E}_j^{\text{img}}} \text{Att}_{qk} \tag{8}$$

$$\sigma_{ij} = \sqrt{\frac{1}{|\mathcal{E}_i^{\text{img}}||\mathcal{E}_j^{\text{img}}|} \sum_{q \in \mathcal{E}_i^{\text{img}}} \sum_{k \in \mathcal{E}_j^{\text{img}}} (\text{Att}_{qk} - \mu_{ij})^2} \tag{9}$$

$$\text{H}_{ij}^{\text{img-img}} = \{(q, k) \mid \text{Att}_{qk} > \mu_{ij} + \beta_2 \cdot \sigma_{ij}, q, k \in \mathcal{I}\} \tag{10}$$

$$\text{Att}'_{qk} = \begin{cases} -\infty & \text{if } q \in \mathcal{E}_i^{\text{img}}, k \in \mathcal{E}_j^{\text{img}}, \text{ and } (q, k) \in \text{H}_{ij}^{\text{img-img}} \\ -\infty & \text{if } q \in \mathcal{E}_i^{\text{img}}, k \in \mathcal{E}_j^{\text{txt}} \\ \alpha \cdot \text{Att}_{qk} & \text{if } q \in \mathcal{E}_i^{\text{img}}, k \in \mathcal{E}_i^{\text{txt}} \\ \text{Att}_{qk} & \text{else} \end{cases} \tag{11}$$

**Notation:** $\mathcal{I}$: set of all image tokens indices, $\mathcal{E}_i^{\text{txt}}$: text tokens of entity $i$, $\alpha$: score scaling factor. $\beta_1, \beta_2$: constant std multipliers.

The cases in 11 correspond to:

- **First case:** Image-to-Image Leakage Suppression
- **Second case:** Image-to-Text Leakage Suppression
- **Third case:** Self-Identity Strengthening

## C.3 LATENCY ANALYSIS

*DeLeaker* introduces computational overhead due to the calculation of entity masks and the statistical analysis (mean and standard deviation) of attention values required to identify target locations for mitigating leakage. To quantify this overhead, we measure the average generation time over 50 instances using FLUX with 20 inference steps. We compare our approach against RAG-Diffusion and 3DIS (Table 9). *DeLeaker* generates an image in 7.68 seconds. While this represents a 44.1% increase in latency compared to the baseline (5.33 seconds), it remains significantly faster than competing methods such as 3DIS (9.43 seconds) and RAG-Diffusion (21.71 seconds).

# D    QUALITATIVE COMPLEMENTARY RESULTS

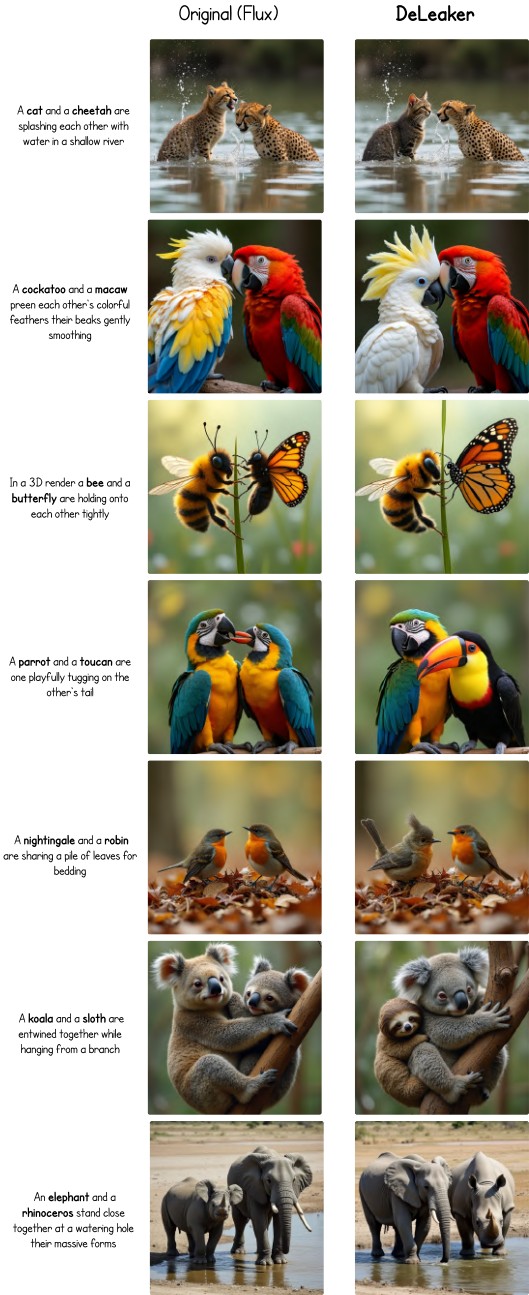

Figure 9: Qualitative Examples - FLUX-based DeLeaker.

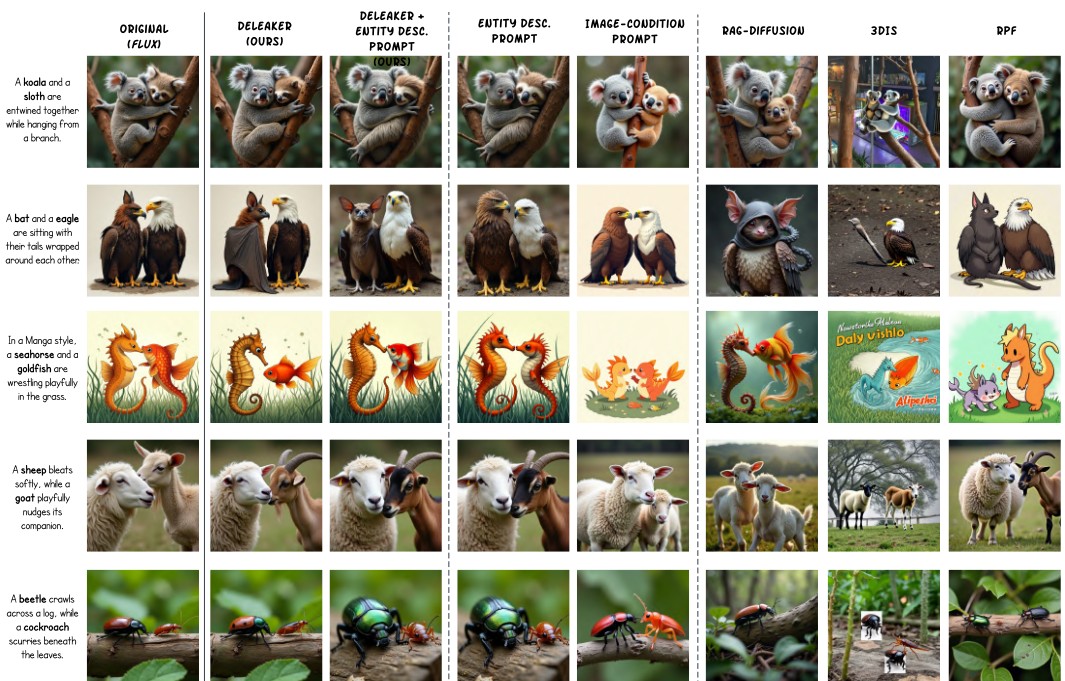

Figure 10: Qualitative comparison across baselines. FLUX-based *DeLeaker*.

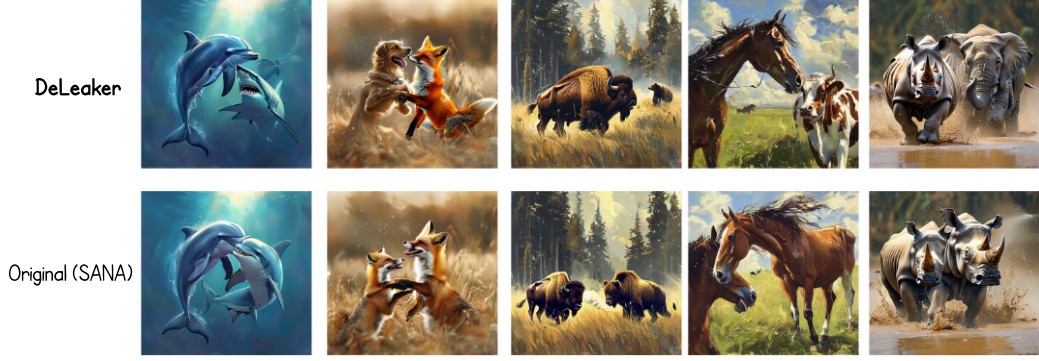

Figure 11: Qualitative Examples. SANA-based DeLeaker.

ORIGINAL          DELEAKER
(WITHOUT SEMANTIC LEAKAGE)

A **dolphin** leaps out of the water while a **butterfly** flutters near the shore

A **zebra** is grazing in the grass while an **owl** is flying above

A **lion** roars from a rocky outcrop while a **penguin** waddles across the ice

An **elephant** trumpets loudly while a **spider** weaves its web in a nearby tree

A **bear** catches **fish** in a rushing river while a **parrot** perches on a tropical

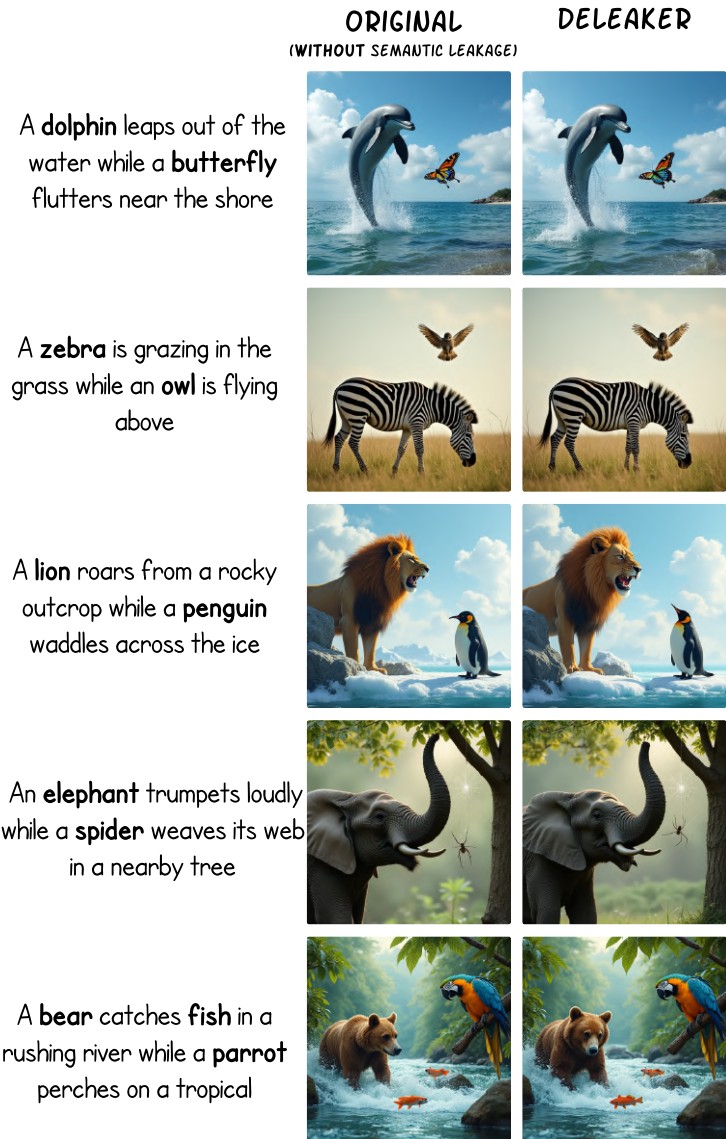

Figure 12: Qualitative Examples of cases when original images do not present semantic leakage. Original images are on left and *DeLeaker* images are on right. *DeLeaker* preserve the image content and quality.

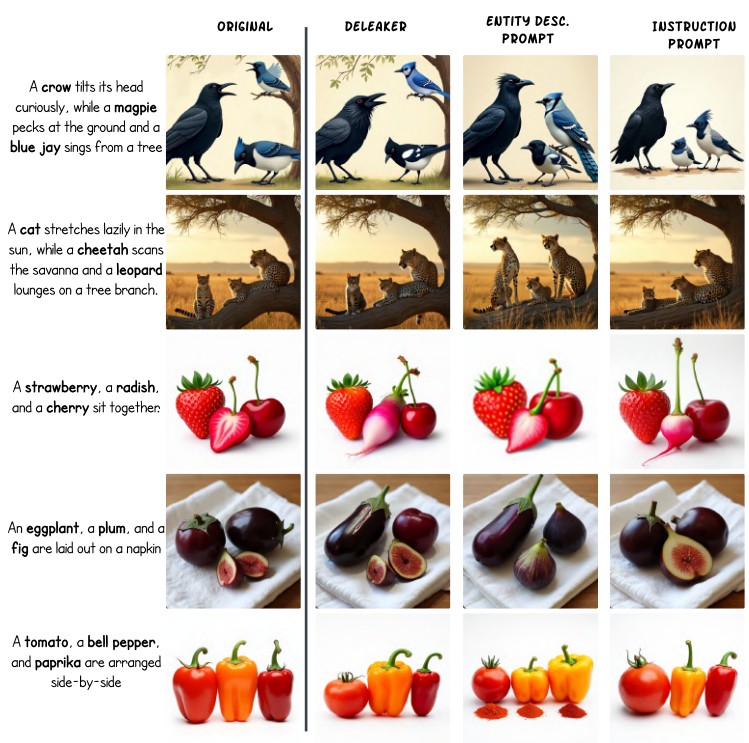

Figure 13: Qualitative Examples of Triplets subset (with original image without entity counting issues). Examples across best performing prompt-based baselines.

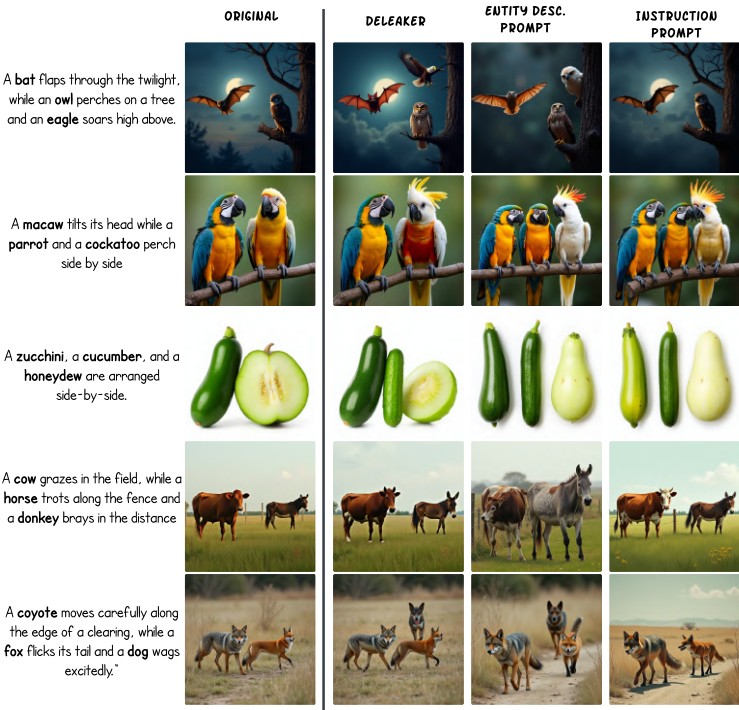

Figure 14: Qualitative Examples of Triplets subset (with original image without entity counting issue: Missing Entity). *DeLeaker* mitigates the leakage in some cases while challenged in others creating the missing third entity. Examples across best performing prompt-based baselines.

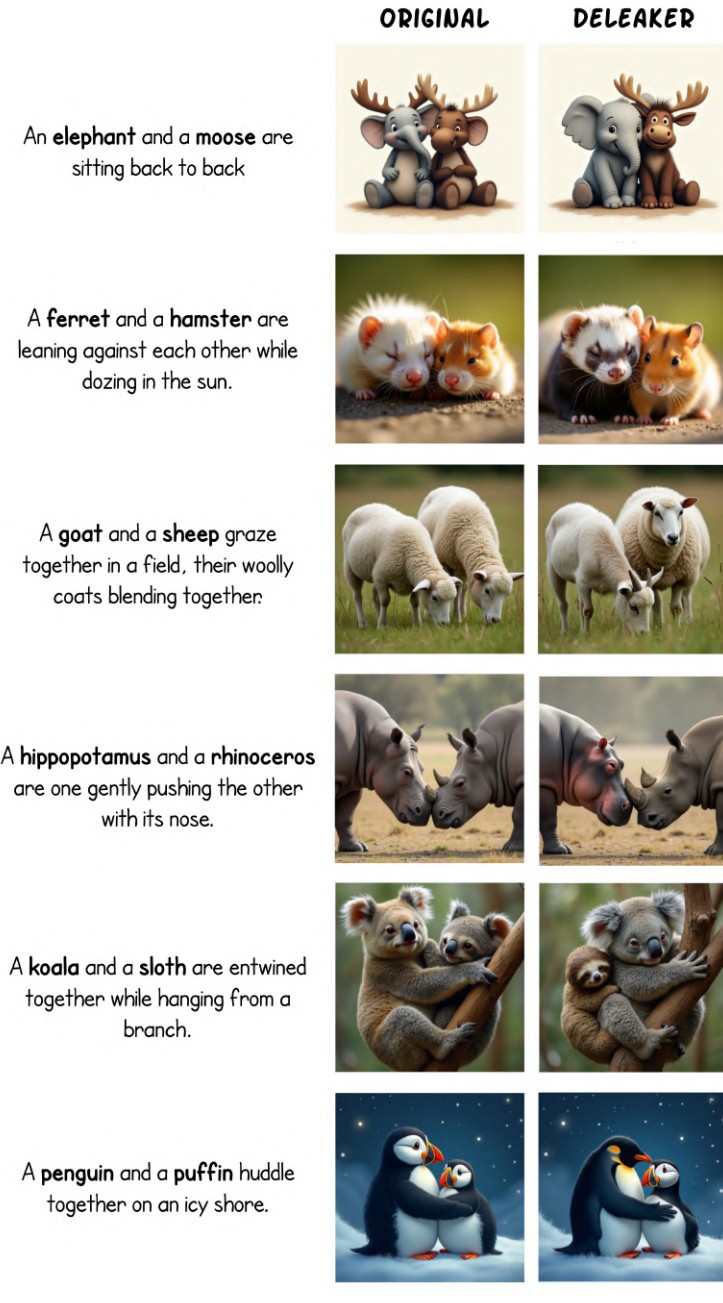

ORIGINAL    DELEAKER

An **elephant** and a **moose** are sitting back to back

A **ferret** and a **hamster** are leaning against each other while dozing in the sun.

A **goat** and a **sheep** graze together in a field, their woolly coats blending together.

A **hippopotamus** and a **rhinoceros** are one gently pushing the other with its nose.

A **koala** and a **sloth** are entwined together while hanging from a branch.

A **penguin** and a **puffin** huddle together on an icy shore.

Figure 15: Qualitative examples from the animal interactions subset. Although *DeLeaker* is effective at mitigating semantic leakage, the suppression of high attention values can sometimes remove desired semantic information, leading to compromised interactions. For instance, as shown in the third row, while *DeLeaker* correctly generates the distinct animals (a goat and a sheep), the joint grazing action is partially lost, resulting in only the goat grazing. Similarly, in the last row, while *DeLeaker* successfully mitigates leakage from the puffin to the penguin, the mutual huddling interaction is reduced to a one-sided action where only the penguin hugs the puffin. We hope that future research will address this limitation to eliminate leakage in a more fine-grained manner, while preserving intended interactions.

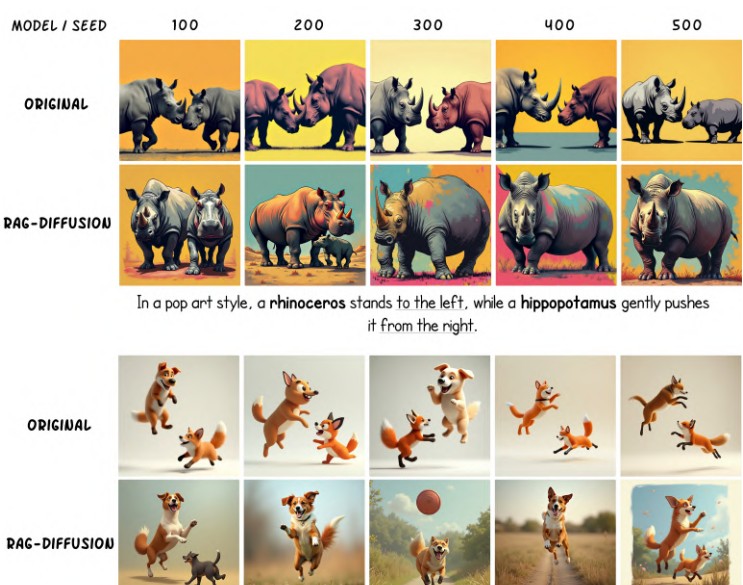

Figure 16: Qualitative examples of explicit location instructions with RAG-diffusion. Each prompt includes an explicit location (such as "to the left") of an entity. We provide generations for five seeds. RAG-diffusion fails to mitigate leakage for these prompts. These results are consistent with those observed in the SLIM dataset.

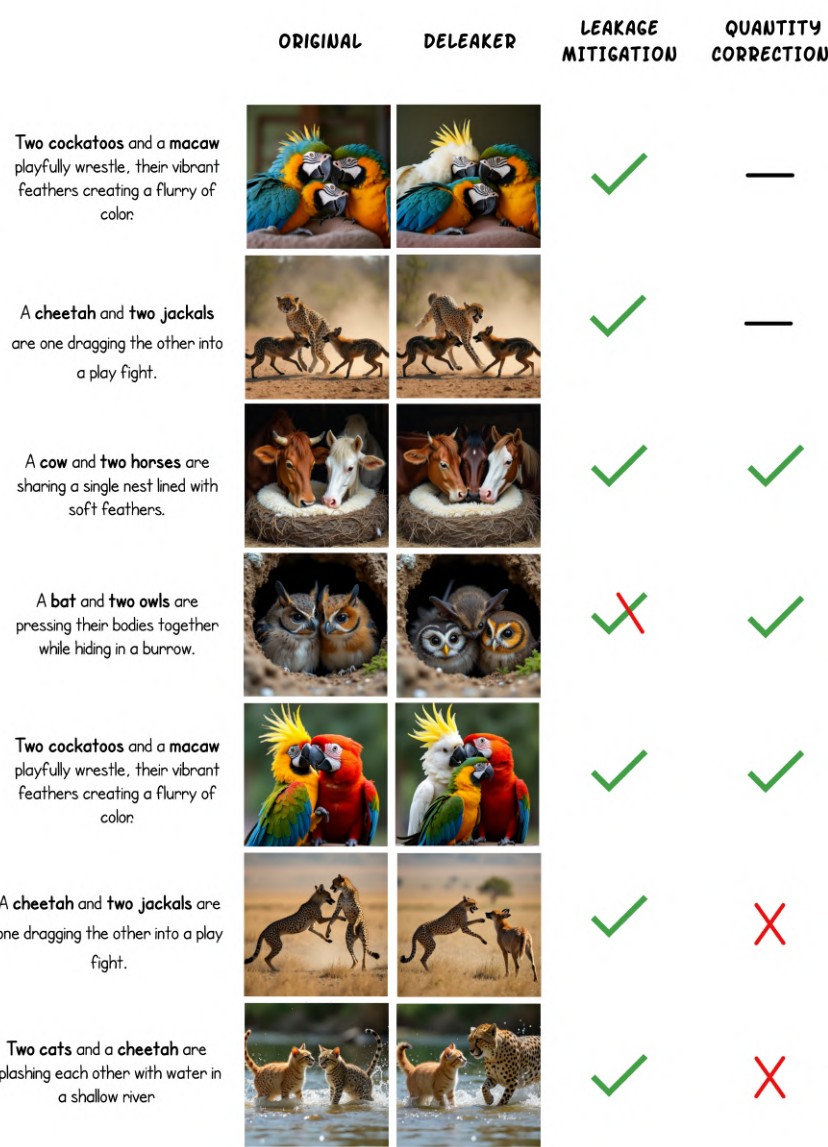

Figure 17: Qualitative results of leakage mitigation versus quantity correction across triplets with identical entities. This subset (additionally to SLIM subsets) includes prompts that specify two identical entities and one additional entity. We observe the following trends: when three entities are generated with leakage (rows 1–2), *DeLeaker* effectively removes the leakage. In many cases, only two entities are initially generated. In such instances, *DeLeaker* partially mitigates the leakage while corrects the quantity (rows 3–5), whereas in other cases (last two rows), the leakage is mitigated but the incorrect number of entities persists.

# E QUANTITATIVE COMPLEMENTARY RESULTS

Table 7: **Human Evaluation Results.** Conducted on MTurk over 60 randomly selected samples across six baselines, with three annotators per task. Aggregation was performed using majority vote, with the median used in case of ties. The table reports the distribution of semantic leakage mitigation, categorized by direction and magnitude of change. Spearman correlation of *0.432* with p-value *<0.001* with the corresponding automatic evaluation (see Appendix Table 8).

| Model | Human Evaluation: Leakage Mitigation (Distribution) | | | | | |
|---|---|---|---|---|---|---|
| | *Visualization* | *Improvement* | | | *Degradation* | |
| | | Major ↑ | Minor ↑ | No Change | Minor ↓ | Major ↓ |
| RAG-Diffusion | | 3.57% | 17.86% | 21.43% | 21.43% | 35.71% |
| RPF | | 5.26% | 22.81% | 26.32% | 21.05% | 24.56% |
| 3DIS | | 5.00% | 16.67% | 16.67% | 21.67% | 40.00% |
| QwenFLUX | | 0.00% | 11.67% | 20.00% | 36.67% | 31.67% |
| Ent. Desc Prompt | | 16.13% | 45.16% | 14.52% | 17.74% | 6.45% |
| DeLeaker | | 13.56% | 54.24% | 25.42% | 6.78% | 0.00% |

Table 8: **Automatic Evaluation Results.** Proportions computed over all user study samples (60)

| Model | Automatic Evaluation: Leakage Mitigation (Distribution) | | | | | |
|---|---|---|---|---|---|---|
| | *Visualization* | *Improvement* | | | *Degradation* | |
| | | Major ↑ | Minor ↑ | No Change | Minor ↓ | Major ↓ |
| RAG-Diffusion | | 16.07% | 3.57% | 10.71% | 3.57% | 66.07% |
| RPF | | 17.54% | 5.26% | 17.54% | 28.07% | 31.58% |
| 3DIS | | 34.48% | 6.90% | 8.62% | 12.07% | 37.93% |
| QwenFLUX | | 16.95% | 6.78% | 11.86% | 20.34% | 44.07% |
| Ent. Desc Prompt | | 36.21% | 8.62% | 29.31% | 10.34% | 15.52% |
| DeLeaker | | 53.57% | 7.14% | 16.07% | 12.50% | 10.71% |

Table 9: Average Generation Times for Different Methods, calculated on a subset of 50 instances.

| Method | Average Generation Time (s) |
|---|---|
| Original | 5.33 |
| RAG-Diffusion | 21.71 |
| 3DIS | 9.43 |
| **DeLeaker** | **7.68** |

## E.1 SANA

The different designs of FLUX and SANA are highly relevant to studying semantic leakage. FLUX combines T5-XXL (Raffel et al., 2020) and CLIP (Radford et al., 2021) encoders, whereas SANA replaces them with Gemma-2 (Team et al., 2024b) and incorporates linear attention in its DiT backbone. These components are crucial, as both the text encoder and attention mechanism dictate how unintended semantic content propagates across modalities.

We evaluate *DeLeaker* effectiveness at mitigating semantic leakage using our human-verified Sana dataset. Since prompt-based baselines have been shown to be more effective for reducing semantic leakage than layout-based methods, and as no implemented layout-based methods are available for Sana, we compare *DeLeaker* against two prompt-based baselines: the instruction prompt and the entity description prompt. The results, presented in Table 10, show that *DeLeaker* is highly effective on the Sana model. It achieves a 64% improvement in leakage mitigation with only a 15% performance degradation, yielding a 49% net improvement. This performance significantly outperforms the instruction baseline. The entity description prompt, however, achieves much better

results due to the additional description of the entities in the prompt, resulting in a score that is only slightly behind *DeLeaker*. We attribute these results to SANA's use of the Gemma model as its text encoder, leading to superior performance on the prompt description baseline. To see whether *DeLeaker* can achieve even better results using this information, we test *DeLeaker* with the entity descriptions. The results are even stronger: *DeLeaker* gains an additional 14% improvement, resulting in a 78% improvement in leakage mitigation and only a 15% degradation, beating the entity description baseline significantly.

Table 10: **SANA Main Results.** Distribution of semantic leakage mitigation across models, categorized by direction and magnitude of change. Arrows (↑ or ↓) indicate the improvement direction. Evaluated on 368 samples, filtered from SLIM large scale with SANA model images.

| Model | Leakage Mitigation (Distribution) | | | | | | Preservation | |
|---|---|---|---|---|---|---|---|---|
| | Visualization | Improvement | | | Degradation | | VQAScore ↑ | LPIPS ↓ |
| | | Major ↑ | Minor ↑ | No Change | Minor ↓ | Major ↓ | | |
| Instruction Prompt (SANA) | | 21.45% | 11.07% | 40.14% | 11.76% | 15.57% | 0.75 | 0.33 |
| Ent. Desc. Prompt (SANA) | | 56.55% | 7.59% | 20.00% | 5.86% | 10.00% | 0.72 | 0.70 |
| DeLeaker (SANA) | | 55.36% | 8.65% | 17.30% | 5.54% | 13.15% | 0.79 | 0.35 |
| DeLeaker With Ent. Desc. (SANA) | | 66.55% | 12.07% | 5.52% | 4.83% | 11.03% | 0.73 | 0.69 |

## E.2 MULTIPLE ENTITIES

To evaluate *DeLeaker* effectiveness with more than two entities, we tested it on two distinct subsets: one featuring prompts including three distinct animals and another containing prompts of three vegetables or fruits. We compared *DeLeaker* performance against two prompt-based baselines: the Instruction Prompt and the Entity Description Prompt. The results, summarized in Table 11, clearly show that *DeLeaker* outperforms both baselines in both the animal and the fruit & vegetable sets.

We observed that *DeLeaker* performance was notably higher on the fruits & vegetables dataset. This is likely because *DeLeaker* is better equipped to handle the generation of duplicate entities, an issue prominent in that particular subset. Its strength lies in a smoothing mechanism across steps and across image tokens, which effectively resolves extra objects that arise from mask duplication. Conversely, the model struggled more with the animal dataset, where the primary challenge was missing entities. *DeLeaker* is less adept at handling this issue because of its design; it cannot create a new mask for an entity if one was not formed in the early stages from the attention maps. Overall, *DeLeaker* is an effective for scenarios with multiple entities, particularly when correcting for duplicates, but future work could focus on improving its performance in cases where entities are missing.

Table 11: **Results on Animal and Fruits & Veg Triplet Subsets (FLUX):** We evaluate leakage mitigation on the triplets subsets across the best performing prompt-based baselines (based on the results on the pair subsets). The main scores represent the percentage of samples labeled as Mitigation (Major/Minor), No Change, or Degradation (Major/Minor), summarized by a stacked bar visualization. These are presented alongside Preservation metrics (VQAScore and LPIPS). Arrows (↑/↓) indicate the desired direction for improvement for each metric.

| Subset | Model | Semantic Leakage | | | | | | Preservation | |
|---|---|---|---|---|---|---|---|---|---|
| | | Visualization | Mitigation ↑ | | | Degradation ↓ | | VQAScore ↑ | LPIPS ↓ |
| | | | Major | Minor | No Change | Minor | Major | | |
| **Animal Triplets** | *Instruction Prompt* | | 39.66% | 3.45% | 19.83% | 0.86% | 36.21% | 0.67 | 0.45 |
| | *Ent. Desc Prompt* | | 38.79% | 2.59% | 13.79% | 6.90% | 37.93% | 0.67 | 0.49 |
| | *DeLeaker* | | 43.97% | 7.76% | 19.83% | 8.62% | 19.83% | 0.70 | 0.25 |
| **Fruits & Veg Triplets** | *Instruction Prompt* | | 35.06% | 10.34% | 15.52% | 3.45% | 35.63% | 0.67 | 0.41 |
| | *Ent. Desc Prompt* | | 51.72% | 8.05% | 8.05% | 7.47% | 24.71% | 0.67 | 0.46 |
| | *DeLeaker* | | 59.20% | 6.90% | 9.77% | 4.02% | 20.11% | 0.70 | 0.34 |

### E.2.1 MULTIPLE ENTITIES: ENTITY COUNTS ANALYSIS

An *entity counts* error in image generation happens when the T2I model fails to create the correct number of entities or items specified in the text prompt. For instance, a prompt asking for "a photo

of a dog and a cat" might incorrectly generate an image showing for example only one dog or two dogs and a cat (Binyamin et al., 2025). This phenomenon signals a failure to maintain alignment between text and image. In many cases, we observe that missing or additional entities are related to severe semantic leakage. This can happen when an entity "disappears" due to leakage, or when two entities fuse into one, creating a blended entity with features from both. Alternatively, a T2I model can generate an additional entity, which complicates the attention relationships among all entities and increases the chance of semantic leakage.

To enrich our analysis, we supplement the main SLIM dataset with an additional set of 222 samples (Table 12). This new subset was specifically filtered to include images with entity count errors, that is, where entities are either missing or added relative to the prompt. The counting is done by prompting Gemini 1.5 pro. Our goal is to use this subset to investigate the link between semantic leakage and these counting errors. We achieve this by assessing whether leakage mitigation techniques also correct the number of entities in these images.

Table 12: **Entity Counts Subset.** This table shows the number of images with missing or extra entities. This additional subset contains 222 samples.

| Group | Subset Name | Additional Entities (Extra) | Missing |
|---|---|---|---|
| Pairs | Animal Pairs | 5 | 9 |
| | Animal Pairs (Interaction) | 3 | 9 |
| | Animal Pairs (Interaction + Style) | 5 | 11 |
| | **Total Pairs = 42** | **13** | **29** |
| Triplets | Animal Triplets | 12 | 116 |
| | Fruit & Vegetable Triplets | 25 | 27 |
| | **Total Triplets = 180** | **37** | **143** |

Based on Table 12, we first observe that entity count errors become more frequent as the number of entities in a prompt increases. The FLUX base model exhibits a notable bias: it tends to generate fewer animals than requested but adds extra items in the fruit and vegetable subset. We hypothesize this bias originates from the training data, where fruits and vegetables are often depicted in groups, while animals are more commonly shown individually.

Figure 18 and Tables 12 and 13 present the results for the entity counts subset, focusing on the pairs subsets and triplets in SLIM, respectively. The results are shown in the form of transitions, tracking the entity count state (missing, same, or extra) from the original image to the candidate image. Figure 18a isolates only the successful transitions (highlighted in green columns of Table 18b, where the model correctly adjusted the number of entities).

When analyzing the baselines on images with the successful entity count, layout-based methods show divergent behaviors: RAG-Diffusion tends to omit entities (56% of cases), whereas 3DIS and RPF tend to add extra ones (21% and 14%, respectively). In contrast, *DeLeaker* is the most stable, preserving the correct number of entities 97% of the time. For cases with missing entities, 3DIS and *DeLeaker*+Desc are most effective at correcting the error. Conversely, when presented with extra entities, most baselines perform well, successfully omitting the surplus items with success rates ranging from 61% to 100%.

Table 13 focuses on the entity count transitions in the Triplet subsets. *DeLeaker* demonstrates better performance in fixing "Missing" entity cases than "Extra" entity cases. We hypothesize that the reason for this is the method's reliance on the generated entity masks; if an entity mask is mistakenly generated, *DeLeaker* continues to intervene based on this incorrect mask rather than omitting it. This presents an interesting direction for future work.

Focusing on the "Missing" and "Extra" columns in both Table 12 and Table 13, we observe that transitions toward the correct entity count are more frequent than transitions that worsen the error. This suggests that semantic leakage mitigation methods generally have a positive effect on entity count errors. **This finding indicates that semantic leakage is a direct cause of entity count issues.**

To further examine multiple entity scenarios in more complex setups, we present Figure 17, which shows qualitative results comparing leakage mitigation and quantity correction across triplets containing identical entities. These examples, which are additional to the SLIM samples, include prompts that specify two identical entities and one additional entity. We observe the following trends: when three entities are generated with leakage (rows 1–2), *DeLeaker* effectively removes the leakage. In

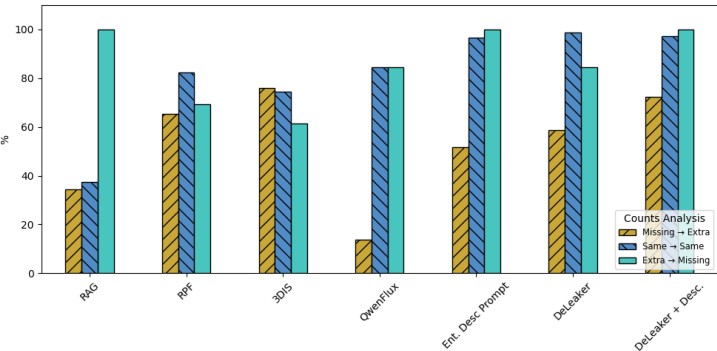

(a) Counts Analysis. Entity quantity transitions between original image → candidate image. The bar graph presents only the successful transitions (green column of the table below).

| Baseline | Original: Missing | | | Original: Same | | | Original: Extra | | |
|---|---|---|---|---|---|---|---|---|---|
| | → Missing ✗ | → Same ✗ | → Extra ✓ | → Missing ✗ | → Same ✓ | → Extra ✗ | → Missing ✓ | → Same ✗ | → Extra ✗ |
| RAG | 0.00% | 65.52% | 34.48% | 56.69% | 37.55% | 5.77% | 100.00% | 0.00% | 0.00% |
| RPF | 0.00% | 34.48% | 65.52% | 3.81% | 82.48% | 13.71% | 69.23% | 30.77% | 0.00% |
| 3DIS | 0.00% | 24.14% | 75.86% | 4.77% | 74.37% | 20.86% | 61.54% | 30.77% | 7.69% |
| QwenFLUX | 0.00% | 86.21% | 13.79% | 11.56% | 84.62% | 3.81% | 84.62% | 7.69% | 7.69% |
| Instruction Prompt | 0.00% | 51.72% | 48.28% | 1.79% | 97.26% | 0.95% | 69.23% | 30.77% | 0.00% |
| Ent. Desc Prompt | 0.00% | 48.28% | 51.72% | 2.26% | 96.79% | 0.95% | 100.00% | 0.00% | 0.00% |
| DeLeaker | 0.00% | 41.38% | 58.62% | 0.48% | 98.81% | 0.71% | 84.62% | 15.38% | 0.00% |
| DeLeaker+Desc | 0.00% | 27.59% | 72.41% | 1.90% | 97.14% | 0.95% | 100.00% | 0.00% | 0.00% |

*✓ Correct model behavior per ground-truth; ✗ Incorrect.*

(b) Entity Quantity Transitions: Percentage of Examples per Baseline.

Figure 18: Visual and tabular analysis of entity count transitions in pairs subsets. The SLIM distribution is: Same: 839, Missing: 29, Extra: 13. (a) Bar graph summarizing the desired transitions across baselines: Same → Same, Missing → Extra and Extra → Missing. (b) Detailed transition matrix showing the percentage of outcomes (Missing, Same, Extra) for each original state.

Table 13: Entity Quantity Transitions for Animal and Fruit & Veg Triplets Subsets: Percentage of Examples per Baseline. Animal Triplets (244 samples: 116 Same, 116 Missing, 12 Extra). Fruits & Veg Triplets: 175 samples: 123 Same, 27 Missing, 25 Extra

| Subset | Baseline | Original: Missing | | | Original: Same | | | Original: Extra | | |
|---|---|---|---|---|---|---|---|---|---|---|
| | | → Missing ✗ | → Same ✗ | → Extra ✓ | → Missing ✗ | → Same ✓ | → Extra ✗ | → Missing ✓ | → Same ✗ | → Extra ✗ |
| **Animal Triplets** | Instruction Prompt | 0.00% | 37.61% | 62.39% | 15.18% | 76.79% | 8.04% | 73.33% | 20.00% | 6.67% |
| | Ent. Desc Prompt | 0.85% | 47.01% | 52.14% | 16.96% | 75.89% | 7.14% | 66.67% | 33.33% | 0.00% |
| | DeLeaker | 0.00% | 63.25% | 36.75% | 5.36% | 83.93% | 10.71% | 26.67% | 66.67% | 6.67% |
| **Fruit & Veg Triplets** | Instruction Prompt | 0.00% | 87.84% | 12.16% | 6.25% | 87.50% | 6.25% | 58.33% | 18.33% | 23.33% |
| | Ent. Desc Prompt | 0.00% | 82.43% | 17.57% | 7.29% | 79.17% | 13.54% | 55.00% | 21.67% | 23.33% |
| | DeLeaker | 0.00% | 79.73% | 20.27% | 1.04% | 64.58% | 34.38% | 28.33% | 36.67% | 35.00% |

*✓ Correct model behavior per ground-truth; ✗ Incorrect.*

many cases, only two entities are initially generated; in such instances, *DeLeaker* sometimes corrects the quantity (rows 3–5) while removing leakage, whereas in other cases (last two rows), the leakage is mitigated but the incorrect number of entities persists.

### E.3 ABLATION STUDY: COMPLEMENTARY RESULTS

Table 14: **Automatic Evaluation Scores of Semantic Leakage Mitigation: Subset Analysis (*De-Leaker*).** The main scores represent the percentage of samples labeled as Mitigation (Major/Minor), No Change, or Degradation (Major/Minor), summarized by a stacked bar visualization.

| Subset | Leakage Mitigation (Distribution) | | | | | |
|---|---|---|---|---|---|---|
| | *Visualization* | *Improvement* | | *No Change* | *Degradation* | |
| | | Major ↑ | Minor ↑ | No Change | Minor ↓ | Major ↓ |
| Animal Pairs | | 31.71% | 10.67% | 40.24% | 6.10% | 11.28% |
| Animal Interactions | | 54.72% | 7.92% | 17.36% | 6.04% | 13.96% |
| Animal Interactions + Style | | 55.87% | 10.53% | 14.17% | 5.26% | 14.17% |

Table 15: *DeLeaker* **Ablation Study.** Configurations are divided into two types: (1) `W/O` rows (top four) represent the removal/addition of a specific component, while (2) `Only` rows (bottom three) isolate each component independently. Absolute scores of are *DeLeaker* are reported, with values closer to the regular configuration of *DeLeaker* (first row) indicating similarity.

| Configuration | *Visualization* | *Improvement* | | *No Change* | *Degradation* | |
|---|---|---|---|---|---|---|
| | | Major ↑ | Minor ↑ | | Minor ↓ | Major ↓ |
| DeLeaker | | 46.07% | 9.76% | 25.36% | 5.83% | 12.98% |
| W/O Image-Image(-) | | 46.31% | 10.12% | 26.67% | 4.29% | 12.62% |
| W/O Image-Text(-) | | 42.98% | 7.62% | 27.98% | 6.07% | 15.36% |
| W/O Image-Text(+) | | 25.00% | 7.98% | 43.93% | 7.02% | 16.07% |
| With Text-Text(-) | | 41.79% | 8.93% | 27.26% | 7.02% | 15.00% |
| Only Image-Image(-) | | 11.90% | 5.95% | 61.90% | 7.86% | 12.38% |
| Only Image-Text(-) | | 25.12% | 8.57% | 47.62% | 5.83% | 12.86% |
| Only Image-Text(+) | | 41.55% | 9.64% | 31.19% | 5.12% | 12.50% |

Table 16: *DeLeaker* **Ablation Study (Relative Change).** Configurations are divided into two types: (1) `W/O` rows (top four) represent the removal/addition of a specific component, while (2) `Only` rows (bottom three) isolate each component independently. Percentage change in semantic leakage mitigation distribution relative to *DeLeaker*. Positive values indicate improvement over DeLeaker, and negative values indicate degradation. Darker hues indicate stronger effect, color-coded as positive and negative .

| Configuration | Relative Change in Leakage Mitigation (% vs. DeLeaker) | | | | |
|---|---|---|---|---|---|
| | *Improvement* | | *No Change* | *Degradation* | |
| | Major ↑ | Minor ↑ | | Minor ↓ | Major ↓ |
| DeLeaker | — | — | — | — | — |
| W/O Image-Image(-) | +0.52% | +3.66% | +5.16% | -26.53% | -2.75% |
| W/O Image-Text(-) | -6.72% | -21.95% | +10.33% | +4.08% | +18.35% |
| W/O Image-Text(+) | -45.74% | -18.29% | +73.24% | +20.41% | +23.85% |
| With Text-Text(-) | -9.30% | -8.54% | +7.51% | +20.41% | +15.60% |
| Only Image-Image(-) | -74.16% | -39.02% | +144.13% | +34.69% | -4.59% |
| Only Image-Text(-) | -45.48% | -12.20% | +87.79% | 0.00% | -0.92% |
| Only Image-Text(+) | -9.82% | -1.22% | +23.00% | -12.24% | -3.67% |

# F  Evaluation and Annotation Protocols

## F.1  SLIM Human-guided Filtering: Human Annotation Protocol for Detecting Semantic Leakage

---

**Human Annotation Protocol**

**Annotation Setup.** Each image in our dataset was evaluated independently by the annotators following a multi-step process. For each original generated image, the annotators followed these steps:

1. **Prompt Review:** Read and understand the textual prompt used to generate the image, with special attention to the entities and their intended differences (e.g., "a horse and a zebra").

2. **Entity Identification:** Identify all relevant entities mentioned in the prompt (e.g., animals, objects, or attributes such as "striped" or "spotted").

3. **Reference Collection:** Use web-based image search engines (e.g., Google Images, Bing) to collect exemplar images for each entity separately. These serve as grounding references for typical visual features of each entity class.

4. **Feature Comparison:** Compare the reference exemplars to identify key distinguishing features between the entities (e.g., color, texture, morphology).

5. **Image Inspection:** Carefully examine the generated image and evaluate the appearance and distinctiveness of each entity.

The full process is illustrated below in Figure 20.

**Labeling Criteria.** Each image was assigned a binary label indicating the presence (positive) or absence (negative) of semantic leakage, based on the following criteria:

*Positive Label (Semantic Leakage Present):*

- **Entity Indistinguishability:** If the entities appear visually indistinct or interchangeable (i.e., they resemble two instances of the same entity class), the image is labeled as containing semantic leakage.

- **Cross-Entity Feature Leakage:** If at least one entity visibly incorporates a feature that is uniquely associated with the other entity (e.g., the spotted pattern of a dalmatian appearing on a golden retriever), the image is labeled positive.

- **Hybridization Effects:** If the image contains a hybrid or fused representation that cannot be clearly attributed to either entity independently, this also qualifies as leakage.

*Negative Label (No Semantic Leakage):*

- **Independent Feature Attribution:** Entities are clearly distinguishable and all major features can be unambiguously attributed to the correct referents.

- **Non-Semantic Artifacts:** Any visual inconsistency that does not reflect semantic leakage, such as color blending with the background, pixelation, blur, rendering artifacts, or lighting inconsistencies, is not considered leakage and is labeled negative.

- **Partial Occlusion or Simplification:** Cases where entities are simplified or partially occluded, but still distinguishable based on remaining cues, are not counted as leakage.

---

Figure 19: Protocol followed by human annotators for assessing semantic leakage in generated images.

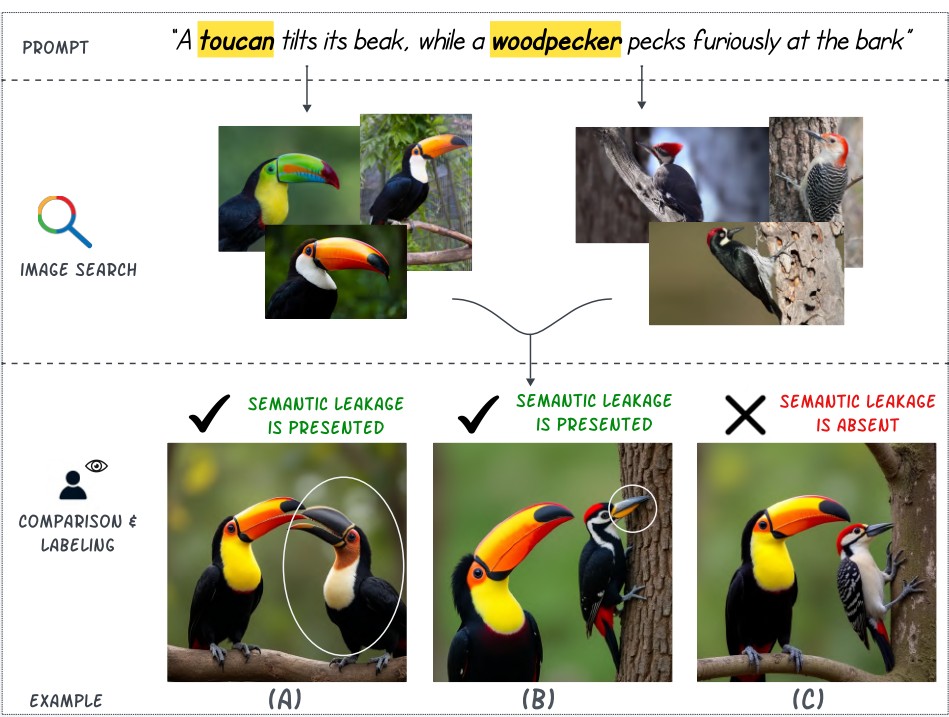

Figure 20: Human Annotation Protocol for Detecting Semantic Leakage. The protocol begins with reading the prompt and identifying the entities involved. Annotators then search for reference images of each entity online and visually compare them to identify features uniquely associated with one entity that appear in 1the other. Examples show (A) significant leakage, (B) localized leakage, and (C) a clean case (without semantic leakage).

### F.2   HUMAN ASSESSMENT OF MITIGATED SEMANTIC LEAKAGE (AMT)

To complement the automatic evaluation and validate its outcomes, we conducted a structured human evaluation using the Amazon Mechanical Turk (AMT) platform. The purpose of this evaluation was twofold: to assess model performance based on human judgment, and to verify the accuracy and robustness of the automatic leakage detection.

Each questionnaire item included a visual figure composed of five elements: two generated images for comparison (the original image suspected of semantic leakage and a baseline image), two reference images (one for each entity, generated by the base (uniintervened) model), and the prompt used to generate the images. The reference images and prompt were provided to help annotators accurately identify the entities and distinguish between them, especially in cases where prior familiarity with the entities could not be assumed. This structure also aligns with the inputs used in the automatic evaluation, allowing for consistent comparison between the two protocols.

Annotators were presented with two questions per item, one for each entity (see Figure 22). For each question, they were asked: "In which image does the [entity] look more typical?" The response was given on a five-point scale, indicating both the chosen image and the strength of preference, such as "Image 1 – strongly" or "Image 2 – slightly." The order of the images was randomized in each instance to mitigate positional bias. Figures 21a and 21b show examples of the visual figure used in the task.

To ensure annotation quality and consistency, annotators were provided with prerequisite guidelines (Figure 23), examples, and a clear definition of typicality. The evaluation covered 60 randomly sampled prompts across six baselines, including *DeLeaker*. Each item was annotated by three independent raters. This resulted in a total of 980 responses. Inter-Annotator-Agreement is moderate, computed for each question (each entity) with an averaged quadratic weighted Fleiss $\kappa$ of 0.52 (0.497 and 0.541 for the two questions).

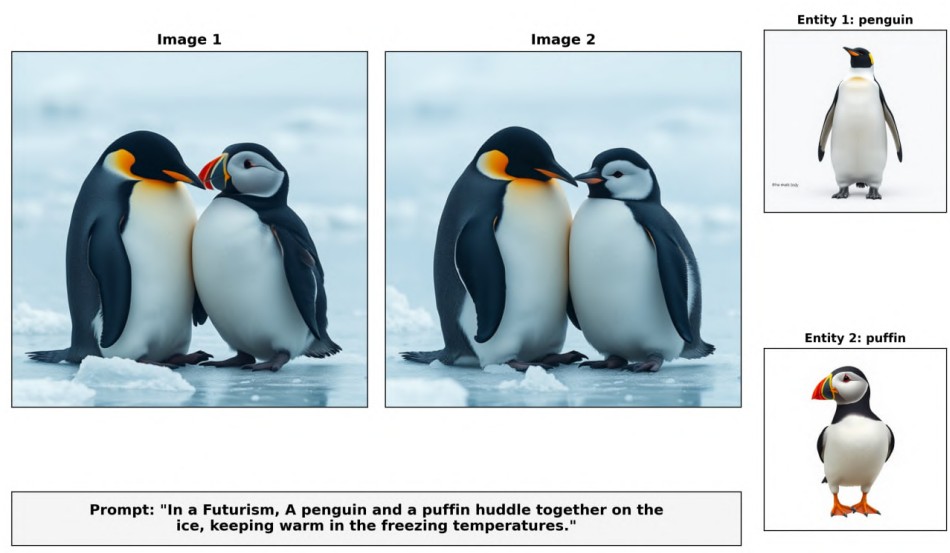

(a) User Study Image Screen: Example 1

(b) User Study Image Screen: Example 2

Figure 21: Two examples of the image screens presented in our AMT user study.

> ### Image Comparison Evaluation Task
>
> **Definition — Typicality:** How usual or expected an entity looks in the image.
> *Example: A tall giraffe is more typical than a short one.*
>
> **Question 1: Entity1 is more typical in:**
> - Image 1 (strong preference)
> - Image 1 (slight preference)
> - Equally typical in both images
> - Image 2 (slight preference)
> - Image 2 (strong preference)
>
> *Use the reference image, your general knowledge, or an online search.
>
> **Question 2: Entity2 is more typical in:**
> - Image 1 (strong preference)
> - Image 1 (slight preference)
> - Equally typical in both images
> - Image 2 (slight preference)
> - Image 2 (strong preference)
>
> *Use the reference image, your general knowledge, or an online search.
>
> **Note:** See the guide for clarifications, examples, and important notes.

Figure 22: The questions in the user study annotation task as appear in the AMT interface for human evaluation.

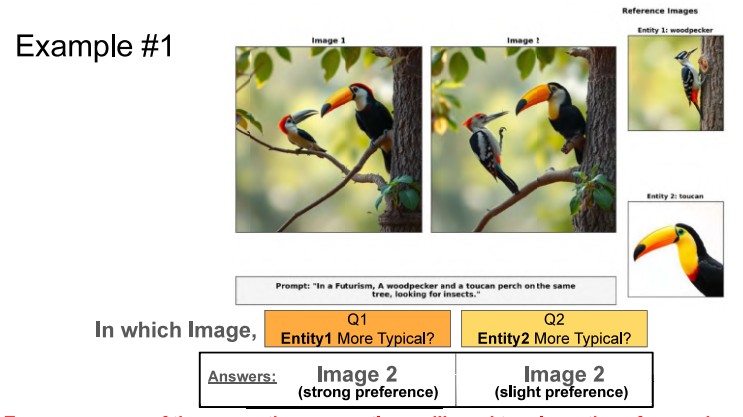

(a) Example guidance for annotators. The guideline here focuses on typical vs. atypical traits.

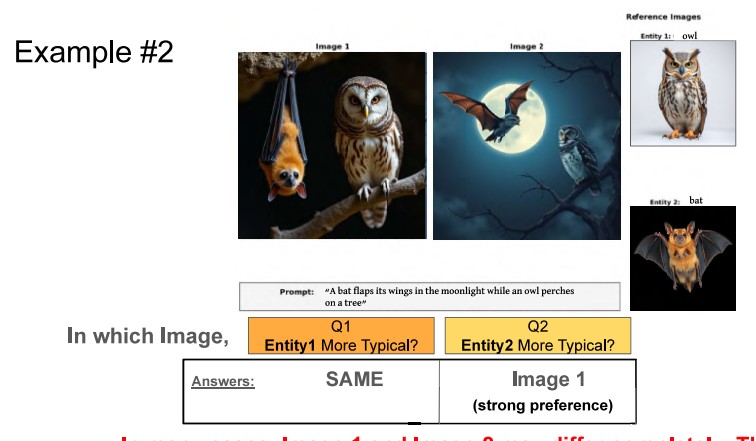

(b) Example guidance for annotators. The guideline here focuses on entity distinction.

Figure 23: Guidance materials shown to annotators before the task, including examples of visual differences relevant for typicality and entity differentiation.

# G   SLIM DATASET CURATION

## G.1   SLIM COMPLEMENTARY DETAILS

Table 17: **SLIM Subsets.** This table details the curation process and the percentage of images exhibiting semantic leakage for each subset. After human-verified filtering, a weighted average of **23% of the initially tested images were found to contain semantic leakage**. The SLIM dataset includes 1,130 samples. Please see Table 12 for the entity count additional set.

| Group | Subset Name | Initial # Images | # Large-Scale Filtering (% of Initial) | Final # After Human Filtering (% of LS / % of Initial) |
|---|---|---|---|---|
| Pairs | Animal Pairs | 1000 | 790 (79%) | 328 (42% / 33%) |
| | Animal Pairs (Interaction) | 1000 | 888 (89%) | 265 (30% / 27%) |
| | Animal Pairs (Interaction + Style) | 1000 | 828 (83%) | 247 (30% / 25%) |
| Triplets | Animal Triplets [FLUX] | 500 | 361 (72%) | 116 (32% / 23%) |
| | Fruit & Vegetable Triplets | 500 | 414 (83%) | 175 (42% / 35%) |

Table 18: **Subsets onSANA.** Human-verified semantic leakage generated with SANA model. This table quantifies the number of images in each subset.

| Group | Subset Name | Final # Images after Human Filtering |
|---|---|---|
| Pairs | Animal Pairs (Classic) | 91 |
| | Animal Pairs (Interaction) | 213 |
| | Animal Pairs (Interaction + Style) | 64 |
| | **Total** | **368** |

Table 19: **Subsets Prompt Templates and Examples:** Each template defines the structure used to generate prompts in a subset of SLIM.

| Subset Name | Description | Example Prompt |
|---|---|---|
| Animal Pairs | Visually similar pairs (from the same breed / family / share similar traits) | "A cow and a horse in a farm" |
| Animal Pairs (Interaction) | Entities perform actions together, share interaction or proximity | "A raccoon is hugging an opossum" |
| Animal Pairs (Interaction + Style) | Entities perform actions together, share interaction or proximity and the same image style | "A watercolor painting of a raccoon dancing with an opossum" |
| Animal Triplets | Visually similar triplets | "A raccoon, an opossum, and a panda playing together" |
| Fruit-Vegetable Triplets | Visually similar triplets | "A bowl containing a strawberry, a tomato, and a cherry" |

Table 20: **Prompts for generating the prompts in SLIM (with GPT-4o).**

| Prompt for Generation | Goal / Subset |
|---|---|
| "From the provided list of animals, please create a list of tuples.  Each tuple should contain a pair of animals that have similar visual features.  For example:  (Cow, Horse), (Goat, Sheep)" | pairs of similar animals |
| "A {animal1} and a {animal2}" | Animal pairs prompt |
| "For each pair you create, write a unique, descriptive sentence.  The sentence must include both animals from the pair, placing them together in a plausible scene." | Animal Pair (Interaction) |
| "Write a unique, descriptive sentence for each pair of animals you create.  The sentence must depict the two animals interacting closely within a plausible scene." | Animal Pair (Interaction) |

**GPT-4o Animals List**   List of additional animals: "Aardvark, Armadillo, Baboon, Beaver, Bongo, Caracal, Cheetah, Chipmunk, Dugong, Elk, Ferret, Gazelle, Giraffe, Guinea pig, Jackal, Llama, Lynx,

Meerkat, Mink, Mole, Moose, Platypus, Pronghorn, Quokka, Rabbit, Skunk, Sloth, Tapir, Tasmanian devil, Walrus, Weasel, Yak, Albatross, Blue jay, Cassowary, Chickadee, Cockatoo, Cormorant, Crane, Cuckoo, Dove, Egret, Falcon, Finch, Hawk, Heron, Ibis, Kingfisher, Kiwi, Kookaburra, Lark, Macaw, Magpie, Mallard, Nightingale, Osprey, Peacock, Pelican, Pheasant, Quail, Raven, Robin, Roadrunner, Stork, Toucan, Vulture, Warbler, Alligator, Anole, Basilisk, Boa, Bullfrog, Chameleon, Cobra, Crocodile, Frilled lizard, Gecko, Gila monster, Iguana, Komodo dragon, Monitor lizard, Newt, Pit viper, Python, Salamander, Skink, Snapping turtle, Terrapin, Toad, Angelfish, Archerfish, Barracuda, Bass, Blowfish, Carp, Catfish, Clownfish, Coelacanth, Cod, Cuttlefish, Eel, Flounder, Guppy, Halibut, Herring, Lionfish, Manta ray, Marlin, Monkfish, Moray eel, Nautilus, Piranha, Pufferfish, Salmon, Sawfish, Scorpionfish, Sturgeon, Swordfish, Tilapia, Trout, Tuna, Wrasse".

**Fruits & Vegtabales List**  Fruits: Banana, Apple, Pear, Grapes, Orange, Kiwi, Watermelon, Pomegranate, Pineapple, Mango. Vegetables: Cucumber, Carrot, Capsicum, Onion, Potato, Lemon, Tomato, Radish, Beetroot, Cabbage, Lettuce, Spinach, Soybean, Cauliflower, Bell Pepper, Chilli Pepper, Turnip, Corn, Sweetcorn, Sweet Potato, Paprika, Jalapeño, Ginger, Garlic, Peas, Eggplant.

**Style List**  In a 3D render, In a Futurism, In a Manga style, In a Pixar style, In a Van Gogh style, In a concept art, In a cyberpunk aesthetic, In a digital painting, In a fantasy style, In a graffiti style, In a minimalist line art, In a neon glow effect, In a pixel art, In a pop art style, In a retro poster design, In a steampunk illustration, In a surrealist painting, In a watercolor painting, In an Art Deco style, In an ink sketch, In an oil painting.

### G.2 AUTOMATIC (NOISY) FILTERING

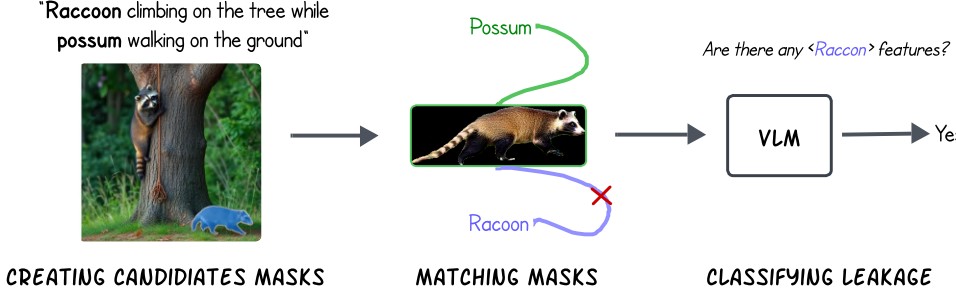

Figure 24: Noisy Automatic Filtering Pipeline Scheme

As illustrated in Figure 24, the automatic noisy filtering pipeline is composed of two main stages. First, we generate segmentation masks for each entity and assign them accordingly. Then, we use a VLM (Gemini) to detect leakage between entity pairs by prompting it with questions such as: "This is an image of a cat. Does it contain features of a dog? Answer yes/no." In evaluating this pipeline, we observed a **non-negligible false-positive rate, which motivated the need for a second-stage filtering process using human-guided filtering.**

**Pipeline Stages.**  The first stage involves generating a specific mask for each entity in an image. The process begins with Grounding DINO (Ren et al., 2024) to detect entity bounding boxes, which are then refined into segmentation masks using the Segment Anything Model (SAM) (Kirillov et al., 2023). This procedure often produces multiple candidate masks for each entity. Trying to increase the chance of an accurate match, we employ the Hungarian Algorithm. This method optimally assigns masks based on a cost matrix derived from CLIP similarity scores (Ramesh et al., 2022), which measure the similarity between each entity's textual token and its candidate masks. This matching step is necessary due to the presence of leakage; when entities share similar features, they may be erroneously classified as the same object during segmentation. After generating the entity masks, the next step is to assess whether a leakage has occurred. This is achieved using Gemini, which evaluates each masked entity to determine if its visual features contain distinctive traits of another entity in the image. The model is prompted with a query of the form: "This is an image of an <opossum>. Does the image contain features unique to a <raccoon>? Answer with yes/no only."

# H  REPRODUCIBILITY AND RESOURCES

## H.1  BASELINES

Table 21: **Model Overview:** Details of the baselines used in our experiments, including their language models, image generation type, and model size. DiT notes Diffusion Transformer.

| Model | Language Model(s) | Image Generator Type | Size |
|---|---|---|---|
| *DeLeaker* FLUX-dev | T5-XXL, CLIP | DiT | 11b |
| SANA | Gemma | DiT | 1.6b |
| RAG-Diffusion, RPF | T5-XXL | DiT | 11b |
| Qwen2vl-FLUX | Qwen2 | DiT | 7b |
| 3DIS-FLUX | T5, CLIP | DiT | |

Table 22: Prompts of the *prompt-based* baselines.

| Baseline | Prompt | Comments |
|---|---|---|
| *instruction* | {prompt}. Each entity maintains its distinct characteristics: entity 1 and entity 2. | |
| *Entity description* | {prompt}. {Entity descriptions}. | where descriptions are generated by Gemini 1.5 pro: Describe the visual appearance of a/an entity in one sentence, focusing only on its unique physical features such as face shape, colors, patterns, and body parts. Keep it short and descriptive. |
| *Image-condition instruction* | The image should depict: {prompt}. Make sure there is no visual leakage between the animals, keep the rest of the image as is. {Image}. | {Image} is the the *original* image by FLUX-dev. |

Table 23: Technical details for the layout-based baselines.

| Baseline | Layout Prior | Strategy | Layout Source | Image Generator |
|---|---|---|---|---|
| RPF (Chen et al., 2024a) | Bounding boxes | local prompts: the entity in the bounding-box | LLM (Gemini 1.5 Flash) | FLUX-dev |
| RAG-Diffusion (Chen et al., 2024b) | Bounding boxes and descriptions | In-bbox attention (referred in the paper as hard binding) + inter-bbox attention (referred as soft refinement) | GPT-4o (originally) | FLUX-dev |
| 3DIS-FLUX (Zhou et al., 2024) | Bounding boxes & depth maps | Depth-map conditioned generation | Stable Diffusion 2.0 (depth map from bbox) (Stan et al., 2023) | FLUX1-depth-dev (Labs, 2024) |

Table 24: Hyperparameters for the layout-based baselines used in our experiments.

| Baseline | Parameter | Value |
|---|---|---|
| RPF | FLUX-dev # steps | 20 |
| | mask inject steps | 10 |
| | base ratio | 0.4 |
| RAG-Diffusion | FLUX-dev # steps | 20 |
| | SR sw split ratio | 0.5;0.5 |
| | HB replace | 2.0 |
| | SR delta | 1.0 |
| 3DIS | Hard control steps | 20 |
| | FLUX-Deph-dev # steps | 20 |
| | SD # steps | 30 |

## H.2 AUTOMATIC EVALUATION PROMPTS

---

**System Prompt**

| Prompt(s) | Comments |
|---|---|
| "As an experienced visual inspector, you will analyze images of entities and provide detailed insights on their visual differences and typicality.  You are sensitive to fine small details and differences." | Defines the overall role of the model as a sensitive visual inspector. |

---

**Step 1.1: Knowledge-Based Extraction**

| Prompt | Comments |
|---|---|
| prompt1 = ["What are the visual appearance differences between {entity1} and {entity2}?  answer in a concise comma-separated list.  For example neck length, head color, eyes shape, etc."] | Extracts visual differences from the model's general knowledge, formatted as a simple comma-separated list. This serves as the first source of information. |

---

**Step 1.2: Image-Based Extraction**

| Prompt | Comments |
|---|---|
| prompt2 = "Based on these images, what are the visual appearance differences between {entity1} and {entity2}?"  + independent entities images | Identifies visual differences by directly analyzing provided images of the two entities. This provides a second, evidence-based source of information. |

## Step 1.3: Integration

| Prompt | Comments |
|---|---|
| prompt3 = ( "List the key visual differences between {entity1} and {entity2} in a bulleted list.  Base your answer on a synthesis of the 'Source 1' and 'Source 2' descriptions provided below.  Each bullet point should concisely compare a single visual feature.  INSTRUCTIONS: 1.  Synthesize Sources:Integrate the key points from BOTH the 'Source 1' and 'Source 2' descriptions.  Do not list the same feature twice or create redundant points. 2.  Highlight Obvious Differences: If either the 'Source 1' or 'Source 2' descriptions explicitly highlight certain features as particularly noticeable or obvious, ensure these differences are prominently featured in your list. 3.  Format:  Start each bullet point with a bolded feature name followed by a colon (e.g., Coat:).  4. Content Structure:After the feature name, first write {entity1}:followed by its description.  Then, on the same line, write {entity2}:followed by its description.  Keep descriptions brief. 5.  Focus:  The list must only contain observable, visual differences.EXAMPLES: Tail Feathers:  Peacock:  Long and iridescent.  Peahen:  Short and brown. Coat:  Zebra:  Black and white stripes. Horse:  Solid or patched color.  Facial Markings:  Red Panda:  White patches on muzzle and eyes.  Raccoon:  Black mask across eyes.  Leg Color:  Flamingo: Pink.  American Coot:  Grey or black. Covering:  Chicken:  Feathers.  Cat: Fur.  Tail:  Red Squirrel:  Long and bushy.  Pig:  Short and hairless. DESCRIPTIONS FOR {entity1} AND {entity2}:  Source 1:  {step1 prompt1 output}.  Source 2:  {step1 prompt2 output}" ) | Merges the text-based differences (from Step 1.1) and image-based evidence (from Step 1.2) into a single, structured, and non-redundant bulleted list. |

## Step 2: Typicality (for entity in entities)

| Prompt(s) | Comments |
|---|---|
| `prompt4 = ("Given the differences between {text_entities}, How visually typical {entity} in this image? (Ignore out-of-frame features.)")  + clean image`

`prompt5 = ("Given the differences between {text_entities}, How visually typical {entity} in this image?  Ignore out-of-frame features.)")  + baseline image` | Evaluates how typical each entity appears in the clean vs. baseline image, ignoring out-of-frame features. Used per entity. |

## Step 3: Ranking (images random order)

| Prompt(s) | Comments |
|---|---|
| `prompt6 = ("Given the independent textual typicality inspection of each animal in each image and the images, overall, how visually typical the {text_entities} in the second image rather in the first image?  (Ignore out-of-frame features.)  Notice that both of the animals should appear in each image.  If one image shows both animals, even if one looks unusual, it will be preferred over an image where one of the animals is missing.  First explain, think step by step.  Finally, rank the overall relative typicality: Rank:  1min (first image with minor prefrence), 1maj (first image with major prefrence), 2min (second image with minor prefrence), 2maj (second image with major prefrence), or 3 (equally typical in both).  First Image Inspection:  {prompt 4 answer for entity 1 and entity 2} Second Image Inspection: {prompt 5 answer for entity 1 and entity 2} First Image:  {clean image}, Second Image:  {baseline image} ")` | |

