# OpenReview forum: "DeLeaker: Dynamic Inference-Time Reweighting For Semantic Leakage Mitigation in Text-to-Image Models"
_ICLR.cc/2026/Conference — ICLR 2026 Poster_

### Official Review · Reviewer_pKUr · 2025-10-22

**Soundness:** 2
**Presentation:** 3
**Contribution:** 2
**Rating:** 4
**Confidence:** 4

**Summary:**

In multiple objects generation tasks, it is well known that content or style leakage occasionally occurs. To address this issue, the proposed paper mitigates leakage in a training-free manner by suppressing or strengthening the model’s attention maps. Furthermore, to evaluate the effectiveness of this mitigation, the authors propose the SLIM Dataset and an automatic evaluation framework. The evaluation framework leverages a vision-language model (VLM) to extract differences between images and uses these as key visual difference metrics.

**Strengths:**

- The method does not require any additional mask nor bounding box, and it can be applied as a training-free manner, which is highly practical. This approach seems general enough to be applied to other diffusion architectures that use attention layers.
- Proposing the SLIM dataset to systematically evaluate the content leakage problem, which has not been thoroughly studied before, and using a VLM-based automatic evaluation framework is a creative and meaningful contribution.
- The method achieves significantly better results than previous approaches in both automatic evaluation and human preference studies.

**Weaknesses:**

- Although the method appears simple yet effective, many of its design choices seem heuristic. For example, the assumption that high attention values correspond to unwanted semantic transfer lacks strong theoretical justification and appears to be chosen empirically. Similarly, the thresholds and coefficients that determine this behavior are treated as hyper-parameters.
- There could be cases where high attention values represent meaningful interactions rather than content leakage. From the visualized results, many images seem to lack strong interactions between objects. More results showing related actions between objects (e.g., playing a game together, sharing food) would strengthen the argument.
- Experiments and comparisons related to the automatic evaluation method are insufficient. For example, how does the proposed evaluation differ from a simpler baseline, such as directly querying a VLM about whether content leakage occurred in the target image?

**Questions:**

- The reviewer concerns if the assumption to judge high attention values as content leakage is too strong. Is there any evidence supporting this claim?
  - Does the method still perform well in scenarios with a larger number of objects? What if two objects are the same animal while the other is not?
- As mentioned in the Weakness section, how can it be demonstrated that the proposed evaluation method is superior to a simpler VLM-based querying approach?
- The previous works performance seems too bad even though they also tackle the same problem. For instance, in the case of RAG-Diffusion, adding some rough location (next to, left/right) may produce much higher-quality results.

---

> ### Author Response · Authors · 2025-11-21
> **Clarifications on Attention Mechanism, Multi-Entity Scenarios, and Baselines**
>
> [Part 1 of 2] Response regarding High Attention for Leakage, and Larger number of Entities
>
> We thank you for your detailed review and for acknowledging the practical, training-free nature of our method, its SOTA performance, and the creative and meaningful contribution of our SLIM dataset and evaluation framework.
> You raised several important points regarding the method's justification, potential failure cases, and the evaluation design. We address these concerns directly below.
> We have already conducted additional experiments for this response, and are currently finalizing further tests which will be shared as a separate comment later.
>
>
> **On the Justification of Using High Attention for Leakage** (Weakness 1, Question 1):
>
> We would like to clarify that the connection between high attention values and unwanted semantic transfer is grounded in the fundamental mechanics of the self-attention operation, rather than being a purely heuristic assumption. Theoretically, when distinct subjects share morphological similarities (e.g., the coarse texture of an elephant and a rhino), their corresponding Query ($Q$) and Key ($K$) projections become aligned in the latent space. This alignment naturally yields high dot-product scores ($QK^T$). By definition, the attention mechanism then aggregates the Value vector ($V$) of the interfering subject into the target’s representation. This mathematical process effectively 'mixes' the features of distinct identities, which constitutes the precise definition of information leakage. Therefore, our design choice to penalize these specific high-attention areas is a direct counter-measure to this feature contamination mechanism.
>
> You raise a vital point regarding **meaningful interactions** (e.g., shading, physical contact, or shared context) which also rely on cross-attention. We acknowledge that blindly suppressing attention could theoretically harm these valid relationships. To address this, we specifically evaluated DeLeaker on the "Animal Interaction" subset of our SLIM dataset. Empirically, we observed no degradation in performance on this subset (Table 3). According to your suggestion, we have added specific qualitative examples of complex interactions, such as animals playing or touching, in Figure 15 of the Appendix to demonstrate that DeLeaker preserves these semantic relationships while preventing semantic leakage.
>
> Regarding **Larger number of Entities** (Question 2), that is a great question. We evaluated this on a dedicated subset combining our animal and fruit triplet datasets (Appendix E.2, Table 9).
> When working with more than two entities, semantic leakage is not the only unwanted phenomenon that arises. In scenarios with entity triplets, we found that it is common to generate extra entities (more entities than the prompt requires) or missing entities (some of the specified entities are not created at all). These issues are exacerbated with larger numbers of entities [1,2] making these cases too noisy for an examination of semantic leakage mitigation. When entities are missing or extra entities are generated, DeLeaker is less effective at correcting the entity count. We acknowledge this as a limitation of our method and hope that DeLeaker can be combined in the future with models that handle counting problems better.
>
> As for the **suggestion of using two entities of the same kind** with a third entity of a different kind, that is a very interesting use case we have not yet explored. Following your advice, we created a set of 100 instances across 10 different prompts, and filtered the once with semantic leakage. We have updated the paper (Appendix, Figure 16) with the qualitative results. We find that Deleaker remains effective at removing leakage in this subset. However, the phenomenon of missing entities (mainly observed in the triplets subset of SLIM) is much more prominent here. In such cases, Deleaker is sometimes able to restore the correct quantity; in other instances, it successfully removes the leakage, but the incorrect number of entities persists (see the last two rows in the table).
>
>
> [1] Dahary, O., et al, (2025). Be Decisive: Noise-Induced Layouts for Multi-Subject Generation. In Proceedings of the Special Interest Group on Computer Graphics and Interactive Techniques Conference Conference Papers (pp. 1-12).
>
> [2] Binyamin, Lital, et al. Make it count: Text-to-image generation with an accurate number of objects. Proceedings of the Computer Vision and Pattern Recognition Conference. 2025.

---

> ### Author Response · Authors · 2025-11-21
> **Evaluation and RAG-Diffusion Baseline**
>
> [Part 2 of 2] Response regarding Evaluation and RAG-Diffusion Baseline
>
> Regarding the **evaluation with a simpler baseline** (Weakness 3, Question 3), our initial approach was to build on open-source models, such as CLIP [3] and BLIP [4]. However, since the similarity metrics in these models often function closer to a "bag-of-words" [5] (understanding which objects are in an image, but often fails to understand how they relate to one another or where they are placed), we employed SAM [6] to automatically generate masks for each entity and attempted to measure the similarity of each entity with the entity prompt. These attempts failed, as the results did not align with human annotations. These experiments are described in detail in Appendix B.2. Unfortunately, no open-source VLM that we tried was able to handle this task of leakage detection. Visual comparison that requires an understanding of small details is very challenging (as described in Appendix B.2). Moreover, our attempts to use closed-source SOTA VLMs such as Gemini [7] on a small scale to detect leakage failed. To validate this claim, we will conduct in the following days a full analysis with Gemini, without our pipeline, and compare the results with human evaluation.
>
>
> Regarding **RAG-Diffusion** (Question 4). We have augmented a subset of SLIM with prompts containing rough locations to evaluate whether RAG-Diffusion performs better at mitigating leakage on this kind of prompts. We are currently running the experiment, and will post another response with the results soon.
>
> We once again thank the reviewer for the constructive feedback, which has helped strengthen our paper. We hope the clarifications above address your concerns regarding the method's theoretical justification and limitations. As mentioned, we are currently finalizing the quantitative comparisons for the vanilla VLM baseline and the RAG-Diffusion spatial prompts; we will share these results in a follow-up comment shortly to fully address your remaining questions.
>
> [3] Radford, A., et al. (2021). Learning Transferable Visual Models From Natural Language Supervision. International Conference on Machine Learning (ICML).
>
>
> [4] Li, J., Li, D., Savarese, S., & Hoi, S. (2023). Blip-2: Bootstrapping language-image pre-training with frozen image encoders and large language models. In International conference on machine learning (pp. 19730-19742) (PMLR).
>
> ‏
> [5] Yuksekgonul, M., et al. (2023). When and Why Vision-Language Models Behave like Bags-Of-Words, and What to Do About It? International Conference on Learning Representations (ICLR).
>
>
> [6] Kirillov, A., et al. (2023). Segment Anything. International Conference on Computer Vision (ICCV).
>
>
> [7] Gemini Team, Google. (2023). Gemini: A Family of Highly Capable Multimodal Models. arXiv preprint arXiv:2312.11805.

---

> > ### Comment · Reviewer_pKUr · 2025-11-26
> >
> > The reviewer appreciates that the authors addressed the reviewer’s concerns, and some of the questions were well understood. However, some of the core concerns remain unresolved, and the reviewer would like to provide additional comments.
> >
> > The main weakness raised earlier was the heuristic strategy of suppressing regions with high attention values. The authors emphasize that similar features become aligned in self-attention, causing high attention score. The reviewer agrees that the high attention scores can occur when two subjects share similar texture or appearance. However, the concern was rather about the assumption that any high-attention region should be interpreted as content leakage. This remains too strong of an assumption. For example, in Figure 1, the elephant’s tusk may have been suppressed simply because its shape resembles the rhino’s horn. In fact, the reviewer believes that Figure 15 shows cases where such suppression does occur: In the third row, both animals were grazing, but after suppression only the goat is grazing. In the last row, although both were huddling, the penguin alone ends up hugging. In Figure 9 as well, the tugging interaction between the parrot and the toucan disappears. These examples suggest that suppression may affect valid interactions.
> >
> > The question regarding multiple-object scenarios was raised because attention becomes more distributed when more objects are present, potentially weakening the proposed suppression mechanism. The question about “two identical objects and one different object” was asked for the same reason. In the provided results, the magpie retains part of the blue jay’s feather shape, and the radish retains the cherry stem. In Figure 16, the leakage around the face of macaw was less effectively mitigated than in the two-object case. This suggests that a single global threshold may not be sufficient and that image-dependent thresholding or adaptive suppression strength might be more appropriate.
> >
> > Regarding the proposed dataset and the use of open-source models, the reviewer is satisfied with the authors’ clarification and detailed explanation.
> >
> > For RAG-Diffusion, the reviewer did not intend to impose a heavy burden. The suggestion was simply to test a few representative prompts (such as those shown in Figure 4) by adding coarse spatial cues like “to the left/right” and checking whether leakage decreases.
> >
> > Overall, the method appears practically useful and addresses the leakage problem in a clever way. The reviewer’s primary concern was that suppressing leakage might unintentionally weaken other aspects of the base model, such as interaction. However, after reviewing the extensive additional experiments, the reviewer acknowledges that although certain behaviors may be weakened, the method reliably improves leakage mitigation, which is the core objective. The reviewer believes that releasing this work to the community would be valuable and will therefore raise the score to marginally accept.

---

> > > ### Author Response · Authors · 2025-12-04
> > > **Closing Remarks to Reviewer pKUr**
> > >
> > > We sincerely thank the reviewer for the thorough engagement and constructive feedback throughout this process. We are encouraged by your assessment that this work is valuable to the community and appreciate your decision to raise the score.
> > >
> > > We acknowledge that our suppression strategy involves a trade-off regarding complex interactions. We have updated the Appendix, Figure 15 caption to reflect this, using the specific examples you identified to illustrate where the assumption may be too strong.
> > >
> > > We have completed the experiment using prompts with spatial cues ("left/right"). As shown in the new Figure 16 (Appendix), RAG-Diffusion fails to mitigate leakage even with these added constraints, confirming that spatial prompting alone is insufficient for this specific issue.
> > >
> > > We believe that with these revisions, the paper now offers a strong and reliable foundation for future work on leakage mitigation and evaluation.

---

> ### Comment · Area_Chair_hAGK · 2025-11-25
>
> Dear Reviewer pKUr,
>
> The authors have responded to your reviews. Please review and provide your feedback and responses.
>
> Best,
>
> Your AC

---

### Official Review · Reviewer_UoR2 · 2025-10-27

**Soundness:** 3
**Presentation:** 3
**Contribution:** 4
**Rating:** 6
**Confidence:** 2

**Summary:**

This paper presents DeLeaker, an inference-time methodology designed to specifically handle cases of semantic leakage in text-to-image models. By directly injecting into the attention mechanism of the underlying generation model, DeLeaker is able to perform light-weight inference-time adjustments which results in more robustly generated images.

In addition to DeLeaker, the paper also presents both a new SLIM dataset, which is specifically curated to evaluate semantic leakage. Separately the authors also provide a set of evaluation pipeline with justification on its validity based on human assessments.

**Strengths:**

The reviewer notes the following strengths:
- The paper is well written with clearly explained motivation for all methods & datasets.
- The presented DeLeaker method is both intuitive and showcases strong performance under human assessment.
- The SLIM dataset marks a very relevant contribution for all future research into semantic leakage.
- Human evaluation is presented showcasing both clear strengths in DeLeaker and justification for the sanity of the automatic evaluations.

**Weaknesses:**

The reviewer notes the following Weaknesses:
- The reviewer finds some concern with the reliance on attention maps to reliably localize entities in the DeLeaker method. In particular, the reviewer would like to see specific evaluation on cases where attention miss-localization occurs.
- The SLIM dataset could also be expanded, given semantic groupings are limited to animals and fruits.
- The paper contains multiple components (DeLeaker, SLIM, evaluation pipeline) that sometimes overwhelms & detract from its core contributions.

**Questions:**

As noted in the weaknesses, the reviewer’s main concern remains the reliance on attention maps for localization. It would be helpful if the authors could share any additional thoughts or clarifications on this point.

---

> ### Author Response · Authors · 2025-11-21
> **Attention Reliability, Dataset Expansion, and Core Contributions**
>
> We thank the reviewer for their encouraging feedback. We are particularly glad that you found the paper well-written and the motivation clear, and that you recognized the intuitive nature and strong performance of our DeLeaker method. We also appreciate your acknowledgement of the SLIM dataset as a significant contribution to future research.
>
> Regarding the **reliance on attention maps for localization**, we acknowledge that DeLeaker’s performance is contingent on the quality of the cross-attention maps. However, the reliability of these maps for localization is well-established in recent literature. For example, Prompt-to-Prompt [1] leverages attention-based localization to control spatial layout. Similarly, What the DAAM [2] demonstrates that aggregating attention across diffusion steps yields reliable attribution between text tokens and the image regions they affect. Regarding the spesific case where attention miss-localization occurs, we have added such case and a discussion
>
>
> Regarding the **dataset expansion**, we appreciate this valuable suggestion. The SLIM dataset was intentionally designed to cover two highly distinct semantic domains (animals and fruits) precisely to demonstrate the efficacy and generalization of the DeLeaker method across different types of visual entities.
> Crucially, SLIM's strength lies in its human-verified nature, a necessity for a dataset evaluating subtle entity leakage. The data creation process for SLIM requires intensive, hard human tagging to ensure the existence of semantic leakage between entities, a process that is significantly more demanding than standard object detection annotation. We have already human-verified 1,130 samples to create the best and currently only dataset of its kind for measuring entity leakage. Given the substantial effort required for this gold-standard labeling, we view the expansion of SLIM to include other domains (e.g., vehicles, furniture) as a great future work opportunity for the community, building upon the foundational benchmark we have established.
>
> Regarding the paper containing **multiple components**, our intention was to provide a holistic solution, a novel method (DeLeaker), a dedicated benchmark (SLIM), and a necessary evaluation pipeline, to fully address the identified challenge of entity leakage. To enhance clarity, we will strengthen the Introduction and Core Contributions section to more distinctly present DeLeaker as the primary algorithmic contribution, with SLIM serving as the essential framework for its evaluation and the research community. We welcome any specific suggestions from the reviewer on how to re-organize or re-emphasize the material to better highlight the core contribution.
>
> We hope our clarifications fully address your concerns and highlight that DeLeaker is a simple, yet highly effective method that integrates seamlessly with T2I models, leveraging their existing capabilities without the need for external guidance, while still achieving the best semantic leakage mitigation.
>
> [1] Prompt-to-Prompt Image Editing with Cross Attention Control (Hertz et al, 2022)
>
> [2] What the DAAM: Interpreting Stable Diffusion Using Cross Attention (Tang et al, 2022) (ACL best paper)

---

> ### Comment · Area_Chair_hAGK · 2025-11-25
>
> Dear Reviewer UoR2,
>
> The authors have responded to your reviews. Please review and provide your feedback and responses.
>
> Best,
>
> Your AC

---

### Official Review · Reviewer_xvBT · 2025-11-01

**Soundness:** 4
**Presentation:** 3
**Contribution:** 3
**Rating:** 6
**Confidence:** 3

**Summary:**

This paper proposes a training-free, guidance-free inference-time algorithm for mitigating entity-level semantic leakage in text-to-image (T2I) diffusion models.

The method, DeLeaker, constructs per-entity masks from the model’s attention maps, suppresses cross-entity attention to reduce unintended feature transfer, and strengthens self-identity through targeted reweighting of text–image attention.

The authors introduce a VLM-based automatic evaluation pipeline and demonstrate that DeLeaker consistently outperforms all baselines across both automatic and human evaluations.

**Strengths:**

- **Simple and Effective**:
The method is conceptually simple yet highly effective. Its reliance on a single hyperparameter ($\beta_1$) for adaptive attention thresholding is an elegant design choice that highlights its efficiency and robustness. This kind of minimalistic control mechanism is one of the method’s most appealing aspects.

**Weaknesses:**

* **Entity Neglection (minor).**
  A potential limitation arises when the base model’s attention maps fail to properly attend to certain entities.
  For instance, if an entity token receives little or no attention over image patches (“token neglection”), DeLeaker may entirely suppress or ignore that entity, as the method operates strictly on existing attention activations.
  Consequently, its effectiveness strongly depends on the *attention quality* of the underlying base model (e.g., FLUX). In corner cases with poor attention calibration, this could produce undesirable omissions.

* **Fit to Venue (minor).**
  While the paper is solidly executed, it may be more naturally aligned with the *computer vision* community (e.g., CVPR), where readers are more accustomed to attention-based control in generative models.
  That said, as ML and vision research continue to converge, this distinction is becoming less rigid, so I do not consider it a major concern.

**Questions:**

* The method seems potentially sensitive to **token neglect**, i.e., cases where an entity token receives incomplete or biased attention coverage. Is the attention quality in models like FLUX sufficiently strong that this is rarely an issue in practice?

---

> ### Author Response · Authors · 2025-11-21
> **Attention Map Reliance and Entity Neglection**
>
> We appreciate the positive comments on DeLeaker’s simplicity and effectiveness.
>
> Regarding our **reliance on attention maps**, it’s true that DeLeaker relies on the quality of these maps. In cases where insufficient attention is captured between the token and the target, we would not be able to address the leakage. However, when entities are generaled, attention maps are relatively reliable, and many other works use these maps to find the location of entities, including in prompt-to-prompt [1], where the attention maps are used to locate an entity, and use this location to control spatial layout, and What the DAAM [2], that have aggregated attention across diffusion steps if order to create reliable attribution between a text token and the areas it most affected.
>
> **Entity neglection** is a phenomenon that is common in T2I models, particularly when more than two entities are present in the prompt. We observed that the FLUX base model displayed biases (e.g., generating fewer animals or extra fruits/vegetables), which we hypothesize stems from the training data distribution (where fruits are often grouped, and animals are individual). Since no leakage needs to be solved when an entity is neglected, we consider entity neglect to be somewhat orthogonal to the leakage problem we address.
>
> Nevertheless, we examined cases where the original model neglects an entity and observed that activating DeLeaker initiates the entity's attention representation in 36%/20% in animals/fruits triplets of cases, (Appendix E.1, Table 14, “original: missing” column, “-> extra” sub-column). As entity neglect remains an open challenge, we anticipate that future methods addressing it could be combined with DeLeaker to mitigate leakage in these scenarios as well.
>
>
> [1] Prompt-to-Prompt Image Editing with Cross Attention Control (Hertz et al, 2022)
>
> [2] What the DAAM: Interpreting Stable Diffusion Using Cross Attention (Tang et al, 2022) (ACL best paper)

---

> ### Comment · Area_Chair_hAGK · 2025-11-25
>
> Dear Reviewer xvBT,
>
> The authors have responded to your reviews. Please review and provide your feedback and responses.
>
> Best,
>
> Your AC

---

### Official Review · Reviewer_bdJX · 2025-11-01

**Soundness:** 3
**Presentation:** 4
**Contribution:** 3
**Rating:** 8
**Confidence:** 4

**Summary:**

The paper proposes an optimisation-free, lightweight inference time control approach, DeLeaker, which reduces leakage by directly intervening on the model's attention maps. The pipeline dynamically adjusts the attention map weights to suppress cross-entity interactions while strengthening the identity of each entity. The authors also propose a FLUX-generated dataset of semantic leakage consisting of 1,130 images. The DeLeaker pipeline achieves the highest mitigation rate of Major = 46.07% and Minor = 9.76%, with the lowest degradation rates of Major = 12.98% and Minor = 5.83%.

**Strengths:**

The paper writing was clear and easy to follow. I have 2 strengths to highlight:

- Systematic evaluation pipeline and dataset: The creation of SLIM, the first benchmark explicitly targeting semantic leakage, and its associated automatic evaluation framework represent strong empirical contributions.
- Novel inference-time attention intervention: By distinguishing between cross-entity suppression and self-identity strengthening, it achieves a practical balance between leakage mitigation and fidelity preservation

**Weaknesses:**

I have mainly one weakness to highlight in the paper:

-Evaluation dependence on external VLMs: The automatic evaluation framework depends heavily on Gemini outputs and textual reasoning to infer visual differences. While the authors validate it with a human study, this dependence could bias results due to VLM limitations in visual grounding and typicality judgments.

**Questions:**

I require some clarification from the authors in regards to the methodology:

- The exact diffusion step range and sensitivity of Beta1, Beta2, and alpha coefficients to scene complexity are briefly mentioned but not fully justified; a stability analysis across prompt types would help.

- It would also help to clarify whether DeLeaker’s interventions introduce latency overhead or computational trade-offs compared to baseline inference.

---

> ### Author Response · Authors · 2025-11-21
> **Evaluation Dependence on VLMs, Hyperparameters, and Latency**
>
> Thank you for acknowledging the clear writing, the systematic evaluation of the pipeline and dataset, and the novelty of our method.
>
> **Regarding the evaluation dependence on external VLMs**: We agree that an evaluation based on open-source models would be preferable; indeed, this was our initial strategy. We first attempted to build our evaluation on open-source models such as CLIP [1] and BLIP [2]. However, since the similarity metrics in CLIP, as a joint-encoding model, often function closer to a "bag-of-words" [3] (understanding which objects are in an image, but often fails to understand how they relate to one another or where they are placed), we employed SAM [4] to automatically generate masks for each entity. We then attempted to measure the similarity of each entity with the entity prompt. These attempts failed, as the results did not align with human annotations. These initial difficulties led us to experiment also with BLIP2, a generative model, by tracking the probability changes of generating ‘Yes’ as the correct detection of the entity before and after the mitigation. Similarly to CLIP, the BLIP attempts failed as well. These experiments are described in detail in Appendix B.2.
>
> Unfortunately, no open-source VLM we tested was capable of handling the nuances of leakage mitigation, as visual comparison requiring an understanding of fine-grained details is highly challenging. Moreover, even closed-source SOTA VLMs such as Gemini [5] failed at the task when used out of the box, without our carefully designed pipeline. We believe that the strength of our design lies in its specific methodology, which includes breaking the challenge down into smaller, manageable sub-tasks to guide the VLM and providing reference images devoid of leakage. As open-source VLMs improve, they can be integrated in specific parts in our evaluation framework.
>
> Regarding your questions:
> - The same **hyperparameters** were used across all prompt types in the SLIM dataset, including animal pairs, animal triplets, animal interactions, and fruit triplets. This consistency demonstrates the robustness of these parameters across diverse prompts. We also provide a qualitative evaluation of the Mask History, smoothing values, and alpha values in Appendix B.1.
>
> - This is an important point. Thank you for raising it. There is some **latency** introduced by our method due to the calculation of statistics on the attention maps and the interventions. Following your suggestion, we have calculated the average generation time on our servers (A100) using 50 random samples from the SLIM dataset. The vanilla generation of Flux-dev with 20 diffusion steps takes on average 5.33 seconds. Our approach introduces additional latency due to the calculation of the average attention values and applying the masks on the dynamically calculated history is 7.68 seconds. To compare this with other methods, RAG-diffusion for example, takes 21.71 seconds on average for the generation, and 3DIS 9.43 seconds . We have added a table (Appendix E, Table 10) including these results. For your convenience, the table is also provided below.
>
> | Method | Average Generation Time (s) |
> | :--- | :--- |
> | Original | 5.33 |
> | RAG-Diffusion | 21.71 |
> | 3DIS | 9.43 |
> | **DeLeaker** | **7.68** |
>
> [1] Radford, A., et al. (2021). Learning Transferable Visual Models From Natural Language Supervision. International Conference on Machine Learning (ICML).
>
> [2] Li, J., Li, D., Savarese, S., & Hoi, S. (2023). Blip-2: Bootstrapping language-image pre-training with frozen image encoders and large language models. In International conference on machine learning (pp. 19730-19742) (PMLR).
> ‏
>
> [3] Yuksekgonul, M., et al. (2023). When and Why Vision-Language Models Behave like Bags-Of-Words, and What to Do About It? International Conference on Learning Representations (ICLR).
>
> [4] Kirillov, A., et al. (2023). Segment Anything. International Conference on Computer Vision (ICCV).
>
> [5] Gemini Team, Google. (2023). Gemini: A Family of Highly Capable Multimodal Models. arXiv preprint arXiv:2312.11805.

---

> > ### Comment · Reviewer_bdJX · 2025-11-24
> >
> > I went through the rebuttals and feel the arguments were justified. However, one thing I would like to clarify is that the weakness I pointed out referred to the dependence on VLMs. Although having reviewed the failed attempts mentioned in Appendix B.2, and the other challenges posed in automatic evaluation, I understand the limitations and find the authors' efforts commendable in this regard. I stand by my assessment and scoring. Also, I thank the authors for providing the overhead time and hyperparameter clarifications.

---

### Comment · Area_Chair_hAGK · 2025-11-22
**official comment by AC**

Dear Authors and Reviewers,

I would like to thank the authors for providing detailed rebuttal messages on time.

To reviewers: I would like to encourage you to carefully read all other reviews and the author responses and engage in an open exchange with the authors. Please post your first response as soon as possible within the discussion time window. Ideally, all reviewers will respond to the authors, so that the authors know their rebuttal has been read.

Best regards,
AC

---

### Author Response · Authors · 2025-12-04
**Rebuttal and Discussion Summary**

We thank the reviewers for their thorough and constructive feedback. We are encouraged by the general appreciation of our contribution, which is reflected in the positive scores (8, 6, 6, and a revised 6 (from 4) from reviewer pKUr).

Specifically, the reviewers valued the paper’s clarity (bdJX, UoR2), the systematic evaluation framework (bdJX, UoR2), and the novelty of our inference-time approach (bdJX). Reviewer xvBT noted that the method is "simple and effective," while Reviewer pKUr highlighted the practical advantage of not requiring masks or bounding boxes. The SLIM dataset was also appreciated by (UoR2, pKUr).

During the rebuttal, we strengthened the paper based on the reviewers' suggestions. Key updates include providing latency measurements and hyperparameter clarifications (bdJX), and conducting additional experiments on spatial prompts for baselines and multiple-entity scenarios for DeLeaker (pKUr). Further updates include addition of qualitative examples from the interaction subset and a transparent acknowledgment regarding high attention values. During the open discussion, reviewers (bdJX, pKUr) confirmed that the additional experiments and measurements addressed their concerns.

With these revisions addressing the remaining concerns, we believe the paper now presents a robust method for leakage mitigation, offering a strong foundation for future work in this domain.

---

### Meta-Review · Area_Chair_injZ · 2026-01-09

**Summary:**

The paper proposes a method that reduces the leakage of attributes between several objects/entities in an image generated by a text-to-image model. The idea is to identify image regions corresponding to different entities and modifying cross-attention so as to reduce leakage. The paper also introduces a dataset and an evaluation methodology making use of VLMs. The method qualitatively and quantitatively outperforms several baselines.

Based on the reviews, the authors’ rebuttal, and the paper itself, the main strengths and weaknesses are as follows.

Pros:
1. The method and the paper are clear and well presented
2. The method performs well, with small computational overhead
3. The dataset and the evaluation framework are new and useful

Cons:
1. The empirical performance is better than the baselines, but not overwhelmingly great (not a major issue, it doesn’t have to be perfect, the gains are quite large)
2. There are concerns about what happens with the method when attention maps are incorrectly identified. (This would be interesting to analyze more, but overall doesn’t seem like a major issue)
3. The evaluation method depends on VLMs and there is not a lot of analysis on the effect of VLM choice and specific way of using it (the authors highlighted in the rebuttal that they did some analysis)

Overall, the paper proposes an effective method to improve text-to-image generative models with a fairly small computational overhead. This semantic leakage between entities is a real problem in generative models, especially the less powerful ones, so ways to address it are useful. Moreover, the paper proposes a dataset and an evaluation methodology. All this taken together warrants acceptance of the paper for the conference.

**Reviewer Concerns:**

- Over-reliance on attention maps and potentially harming “useful” interactions between entities -> rebuttal addressed this well: yes there’s reliance, but the method still works and doesn’t get rid of all “useful” interactions
- Dependence of evaluation on VLMs -> reasonably addressed, compared to human eval
- The method includes some design decisions that are not necessarily 100% justified -> addressed reasonably well in the rebuttal

**Reviewer Scores:**

One of the reviewers increased the score from 4 to 6. I wouldn’t expect other changes.

---

### Decision · Program_Chairs · 2026-01-26

Accept (Poster)